# Human-induced climate change amplification on storm dynamics in Valencia's 2024 catastrophic flash flood

Carlos Calvo-Sancho [1,2] ✉, Javier Díaz-Fernández[1],
Juan Jesús González-Alemán [3], Amar Halifa-Marín[4,5],
Mario Marcello Miglietta [6], Cesar Azorin-Molina[2], Andreas F. Prein[7],
Ana Montoro-Mendoza[1,8], Pedro Bolgiani [9], Ana Morata[3] & María Luisa Martín[1,10]

Global warming alters the hydrological cycle, increasing heavy rainfall events worldwide. In October 2024, Valencia (Spain) experienced rainfall accumulations in a few hours surpassing annual averages (771.8 mm in 16 h in the official weather station at Turís) and breaking the record for one hour rainfall accumulation in Spain (184.6 mm), resulting in 230 fatalities. Here, we present a physical-based attribution study employing a km-scale pseudo-global warming storyline approach to assess the contribution of anthropogenic climate change. We show that present-day conditions led to a 20% °C$^{-1}$ increase in 1-hour rainfall intensity, exceeding Clausius-Clapeyron scaling. This intensification was driven by enhanced atmospheric moisture from warmer sea surface temperatures, leading to increased convective available potential energy, stronger updrafts, and microphysical changes including elevated graupel concentrations. These results demonstrate that anthropogenic climate change could intensify the occurrence of flash-floods in the Western Mediterranean region: in this particular case, it intensified the 6-h rainfall rate by 21%, amplified the area with total rainfall above 180 mm by 55%, and increased the volume of total rain within the Jucar River catchment by 19% compared to the pre-industrial era. This study highlights the urgent need for effective adaptation strategies and improved urban planning to reduce the growing risks of hydrometeorological extremes in a rapidly warming world.

The attribution of severe convective and extreme precipitation events to anthropogenic climate change (ACC) remains a subject of active scientific debate (IPCC, 2021), due to the historical challenges to determine frequency and intensity changes consequent to limited observational records and mesoscale, nonlinear dynamics. While rising sea surface temperatures (SST) and increased atmospheric moisture content are consistent with thermodynamic expectations of a warmer climate, the role of changes in atmospheric dynamics and storm morphology due to ACC remains uncertain[1,2]. Several studies indicate that breaking Rossby waves plays a critical role in the Subtropics; however, distinguishing between human-induced effects and natural processes remains challenging[3,4]. Nonetheless, the projections of more frequent long-lasting cut-off lows[5] and fewer but more intense precipitation events over the Mediterranean region[6,7] suggest potential changes in convective initiation under higher levels of greenhouse gases (GHG). While these findings may appear contrasting, they reflect the high level of uncertainty surrounding the dynamic response of the atmosphere to anthropogenic forcing. This highlights the need for comprehensive attribution studies that integrate both

thermodynamic and dynamic components to better assess the impact of ACC on extreme precipitation.

Attribution studies for severe rain-flood events are still preliminary, especially concerning physical-based attribution (e.g., storylines using pseudo-global warming simulations). Rapid approaches, such as probabilistic, analog-based or AI-driven models, focus on comparing essential parameters (total precipitation, wind speed, or surface pressure) that can quickly assess the likelihood of ACC playing a role. Initiatives such as ClimaMeter[8] or the World Weather Attribution[9] exemplify how these methods provide a first approach to the way the ACC is influencing record-breaking extreme events. These initiatives suggest the Valencia's floods was twice as likely and 13% more intense because of ACC[10–12]. Physical-based attribution explores the underlying storm dynamics to reveal how ACC may change thermodynamic and atmospheric dynamics, such as complex microphysics cloud processes. Although these detailed analyzes take longer, they offer a more comprehensive understanding of how extreme rainfall events are evolving under a warming climate. Moreover, this approach provides a deeper insight in the sub-daily scale. Most of the attribution studies focus on daily-scale to quantify the contribution of ACC to the intensity, frequency and extent of extreme precipitation events. Conversely, the sub-daily scales where convective processes dominate are still poorly characterized. Herein, we address this lack of knowledge by analyzing sub-daily observations and high-resolution simulations to quantify the contribution of ACC to the Valencia extreme rainfall event.

On October 29th 2024, the eastern Spanish region of Valencia experienced one of the most devastating flash flood events in recent history, with record-breaking rainfall rates (Fig. 1b, c) and cumulative values exceeding annual averages within few hours. The hydrological response was devastating (Fig. 1c), triggering extensive flash floods in the south of Valencia metropolitan area. In addition to the heavy rainfall associated with the convective system, 11 tornadoes and large hail were observed[13]. The event resulted in at least 230 fatalities, extensive damage to infrastructure, and economic losses estimated in several billions of euros[14,15].

The event was driven by a cut-off low over the Iberian Peninsula (Fig. 1a), creating a baroclinic environment where cold air aloft contrasted with warm, moist low-level air advected from the subtropical Atlantic. This strong contrast favored convective instability and the development of quasi-stationary convective systems (Fig. 1d). The cut-off low enhanced convection by inducing upper-level divergence, thereby facilitating intense horizontal moisture advection—like atmospheric rivers—from the Mediterranean Sea and northwestern Africa[16]. This transport increased the supply of humid air and raised the potential for severe storms. This pattern often plays a key role in torrential rainfall events in the eastern Iberian Peninsula[17–21].

This catastrophe highlights the vulnerability of Mediterranean regions to extreme precipitation events, which seems to be exacerbated in recent years by ACC[22–26]. The anomalously high SST in the western Mediterranean, which keeps increasing and reached record levels during summer 2024[27], likely intensified the event through thermodynamic forcing, increasing atmospheric moisture content and enhancing convective instability[28]. This mechanism is consistent with theoretical expectations of a warmer climate following the Clausius-Clapeyron relation, which projects an approximate 7% increase in saturation vapor pressure per degree of warming[29–33]suggested an intensification due to the warming climate in the sub-daily scale precipitation exceeding the

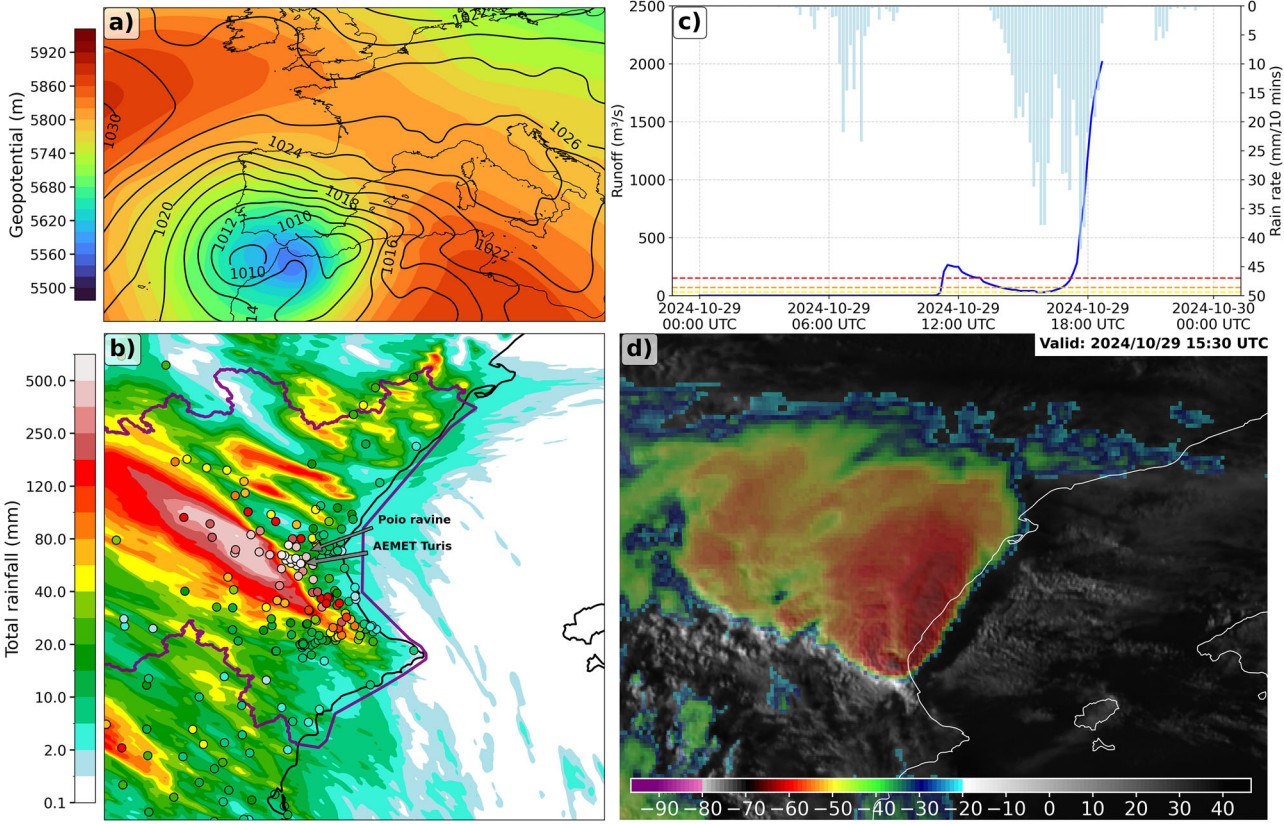

**Fig. 1 | Valencia's deadly floods are the highest impactful climate event in recent Spanish history. a** Geopotential height at 500 hPa (shaded) and sea level pressure (contour) on October 29th, 2024, 12:00 UTC from ERA5. **b** Total rainfall accumulation (24 h) in the Valencia region in the factual simulation (shaded) and observational weather network (scatter). The purple line represents the river Júcar basin. Station locations are indicated. **c** Evolution of 10 min rainfall rate in Turís (light-blue bars) and 5 min runoff (blue line) in Poio ravine. At 19:00 UTC the flash-flood in the Poio ravine destroyed the stream gauging station. The yellow, orange and red lines show the runoff warnings in the Poio ravine. **d** View of the storm from Meteosat Second Generation taken at 15:30 UTC in the IR10.8 μm channel (°C).

Clausius-Clapeyron scaling. However, storyline-based analyzes have pointed out the role of large-scale atmospheric circulation in decreasing precipitation extremes in the Mediterranean region during the cold season[34], while observational records do not show a significant trend in annual precipitation[35]. In contrast, the future European flood risk could notably increase in a warming climate due to higher precipitation intensities and larger area than in present-day climate, increasing the socio-economic impact in Europe[36,37].

In this study we use a convection-allowing model with 1-km horizontal grid spacing to simulate the event in the present-day (factual) and pre-industrial-like (counterfactual) climate conditions. Our analysis follows a storyline approach using the PGW methodology, which enables a physical-based attribution approach considering factual and counterfactual climates. This procedure enables a thorough evaluation of the way ACC has intensified the extraordinary Valencia floods of October 2024 by examining changes in sub-daily rainfall intensity and spatial coverage, changes in moisture content, and shifts in the underlying physical mechanisms governing this extreme rainfall event (further details in Methods section).

## Results

### Changes in rainfall intensity and area

The factual simulation reproduces well the overall spatial distribution of precipitation, although the simulated precipitation field is slightly displaced westward relative to the observations (Fig. 1b). The highest amounts of precipitation are concentrated in a central region, aligning relatively well in terms of location with the station-based observation patterns (Fig. 1b).

Large-scale conditions present some minor differences between factual and counterfactual simulations in the 500-hPa geopotential field (Supplementary Fig. 9). In a preindustrial-like climate, the cut-off low would have been slightly deeper than in the present-day climate, but with a similar location. Regarding the MSLP, there are no notable differences neither in position nor intensity (Supplementary Fig. 9).

Compared with the counterfactual (pre-industrial-like) climate simulations, the factual (present-day) run shows higher precipitation accumulations, higher density of extreme precipitation values and a larger affected area (Fig. 2a, d). The hourly rainfall rates in the factual simulation are consistently higher than those in the counterfactual simulation, particularly for extreme values (Fig. 2a). Additionally, the rate of precipitation change per degree of warming (Fig. 2b) in the factual scenario exceeds the Clausius-Clapeyron scaling (7% °C$^{-1}$) for higher precipitation intensities[31,32,38]. A strong increase in the total area exceeding precipitation thresholds (the total 24-h rainfall percentiles are calculated from the factual simulation) is evident, with the most pronounced changes observed at the highest thresholds (Fig. 2c). For lower thresholds (e.g., <80 mm, below 90th percentile), the increase is more modest and presents lower variability (not shown). In contrast, extreme precipitation events, e.g., percentiles 95th (P95) to 99th (P99), tend to show a substantial expansion in the affected area. There is a median increase of ~50% in the total area exceeding the 180 mm threshold (which is set by the Spanish Meteorological Service as the limit for red warning for heavy precipitation in Valencia's region). These results are consistent with similar previous studies, such as ref. 39, and show an increase in heavy precipitation amount by more than 20% in the future warmer climate. However, our results should be interpreted with caution due to the considerable inter-member variability. Regarding the P99 threshold, 7 out of 15 simulations, as well as the ENS mean, fail to exceed the 300 mm threshold under counterfactual conditions, while it is exceeded in the simulation under factual conditions. This suggests that, at the most extreme levels, the signal may not clearly emerge from the noise (Supplementary Table 2). Nonetheless, this behavior is consistent with the general tendency for higher thresholds to exhibit more uncertain responses, a feature also reported in previous studies[25,39–41]. These increments related to 24-h

total precipitation are also found in previous initialization dates, which demonstrates the robustness of the findings (Supplementary Fig. 10).

In addition, the factual simulation shows a more intense 6-hour accumulated precipitation core (Fig. 2d) compared to the counterfactual pattern, with both higher peak values and a larger affected area. The probability density function (PDF) of 6-h accumulated rainfall further supports this pattern: the ENS mean for the counterfactual runs is significantly lower than the factual distribution, indicating a shift toward more intense precipitation under current climate conditions (Fig. 2d). Finally, the distribution of 6-h accumulated precipitation reveals that the factual simulation not only exhibits higher median precipitation but also a longer upper tail and greater interquartile range, emphasizing the increased variability and frequency of extreme values.

These findings highlight the pivotal role of ACC in increasing the intensity and spatial extent of extreme rainfall events. The substantial increase in precipitation intensity and affected areas aligns with previous studies (e.g.[23,42]), which demonstrate the amplification of extreme rainfall events due to ACC.

### Changes in moisture content and fluxes

The ACC intensified both the magnitude and the spatial extent of the rainfall event. This amplification suggests that enhanced atmospheric moisture may have played a major role in fueling heavier precipitation over the Valencia region, as observed in other cases[23,43]. Our km-scale simulations provide a distinctive opportunity to investigate the role of ACC impacts on the moisture content and fluxes.

The response of extreme convective systems to ACC is strongly influenced by the changes in air temperature, vertical lapse rate and moisture content. The severity of convection can be measured by using the most unstable convective available potential energy (MUCAPE), which has a strong relation with the intensity of mid-latitudes convective storms[23,44–47]. On the one hand, Fig. 3a indicates larger amounts of MUCAPE (peaks of about 2000 J/kg) in the factual simulation, particularly along the Mediterranean coastline and inland in certain areas of the Valencian region. On the other hand, the counterfactual simulations have substantially lower MUCAPE values, with peaks of about 1600 J/kg. This difference is statistically significant ($p$-value < 0.01 by two-sided Mann–Whitney U test). The median MUCAPE increase from counterfactual to factual climate is 22.2%. Moreover, the large amounts of MUCAPE in the factual simulation are closely linked to increased low-level atmospheric moisture content due to warmer air temperatures and increased atmospheric water vapor content (Supplementary Fig. 2)[48–50]. Similar to the saturation vapor pressure, we also expect a theoretical MUCAPE increase close to the Clausius-Clapeyron scaling (Fig. 3a)[51]. This enhancement in MUCAPE for the factual world reflects stronger buoyancy available for convective clouds, enhancing vertical motion and consequently intensifying latent heat release. This intensified latent heating could then lead to rainfall increases exceeding Clausius-Clapeyron rates (Fig. 2b)[23,49,52,53].

Precipitable water (PW) quantifies the total available water vapor and provides a direct measure of the potential to generate precipitation. The factual simulation displays extensive areas of high PW levels, between 25 to 40 mm, denoting the ability to produce high rainfall intensities (Fig. 3b). As expected, the counterfactual simulations show a notably lower PW, with values between 15 and 30 mm (Fig. 3b). The median PW increase from pre-industrial to present-day climate is 11.9%. The differences in PW between factual and counterfactual scenarios are consistent with the water vapor mixing ratio (WVMXR) results: the factual scenario depicts significantly larger amounts of water vapor concentration in comparison to the counterfactual climate (Supplementary Fig. 2). Moreover, the median increase in vertical integral of WVMXR from pre-industrial to current climate is 8.5%. These differences imply that ACC has substantially enhanced the moisture content in the troposphere[28,54–56]. This is consistent with the Clausius-

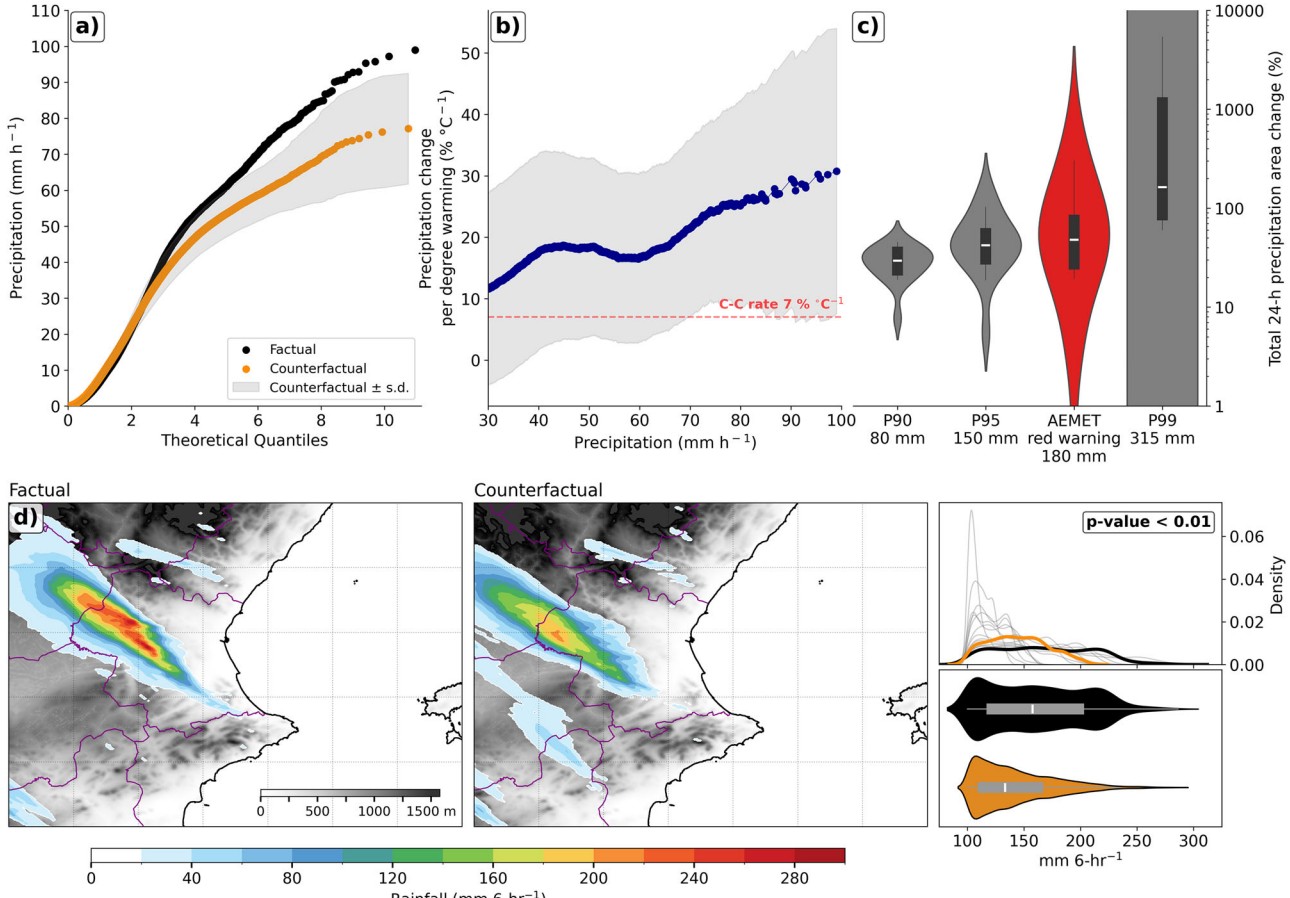

**Fig. 2 | Anthropogenic Climate Change intensified rainfall intensity and increased total rainfall area.** Comparison between the factual and counterfactual simulations during the storm period for: **a** Quantile-quantile plots (empirical quantiles versus quantiles expected under the fitted reference distribution) of 1-h rainfall rate over the Valencia region (domain shown in Fig. 2d) for factual (black) and counterfactual (orange) simulations (± standard deviation is gray shaded). **b** Hourly precipitation extremes scaled by the +1.08 °C warming between factual and counterfactual. The blue line shows % change per °C warming (±standard deviation is gray shaded), and the red dashed line shows the expected Clausius-Clapeyron rate (7% °C⁻¹). **c** Percentage change in the area where the 24-h

accumulated precipitation from the factual simulation exceeds different rainfall thresholds over the Valencia region. Thresholds include the 90th (P90), 95th (P95), and 99th (P99) percentiles, as well as the AEMET red warning threshold (180 mm). The total 24-h rainfall percentiles are calculated from the factual simulation. **d** Spatial distribution for factual (left) and counterfactual (middle) simulations, and (right) PDF and violin plot of 6-h rainfall amount. The PDF includes individual climate model counterfactual runs (gray), the ensemble means for counterfactual (orange), and factual simulations (black). The *p*-value from a Mann–Whitney U test indicates that there is a statistically significant difference between the two distributions.

---

Clapeyron relationship, which indicates that warmer air can hold more moisture content. As a result of recent global warming, atmospheric water vapor content is increasing by about 7% per °C[29–31]. Increased moisture availability leads to greater precipitation intensities and a broader spatial coverage during convective storms, as moist convection processes are sensitive to moisture supply[1,57]. At the same time, this large amount of moisture reduces entrainment dilution of updrafts that promotes higher buoyancy and less evaporative cooling. Therefore, higher moisture availability supports stronger upward velocities and longer-lived cells[58–60]. The positive feedbacks between these mesoscale processes increase precipitation efficiency and can enhance convective organization, intensifying surface rainfall[61–63]. Consequently, the significant increase in PW observed in the factual climate directly contributed to the heightened rainfall intensity during the Valencia event.

In conjunction with the PW and the WVMXR, the water vapor flux into the storm area (WVFlux) is a key variable that controls the rainfall amount, due to the linkage between the mass of water pushed by an intense low-level wind (up to 25 m/s in the present case) and precipitation intensity. The factual climate depicts stronger WVFlux (between surface and 700 hPa) in comparison to counterfactual

simulations (Fig. 3c). The median WVFlux increase from pre-industrial to present-day climate is 8.5%. These significant differences show that in a warmer climate the moisture transport processes are enhanced[64–67], favoring the extreme rainfall event in Valencia. In fact, there are no significant differences in the wind magnitude (Supplementary Fig. 8h); therefore, the main contribution to the differences between factual and counterfactual climates comes from the moisture content and fluxes. Nevertheless, it should be noted that the wind components have not been forced by the climate perturbation signal (see Methods section).

## Changes in physical mechanisms controlling extreme rainfall

The rate of intensification of rainfall extremes under a warming climate depends on storm dynamic mechanisms. Diabatic heating, predominantly latent heat release from condensation processes in clouds, has a notable role in the dynamics and intensification of heavy precipitation events[23,68,69]. The factual world exhibits a spatial distribution of diabatic heating more intense in comparison to the counterfactual scenario (Fig. 4a). The median diabatic heating increase from pre-industrial to present-day climate is +29.5%. This significantly intensified latent heat release denotes reinforced atmospheric convection,

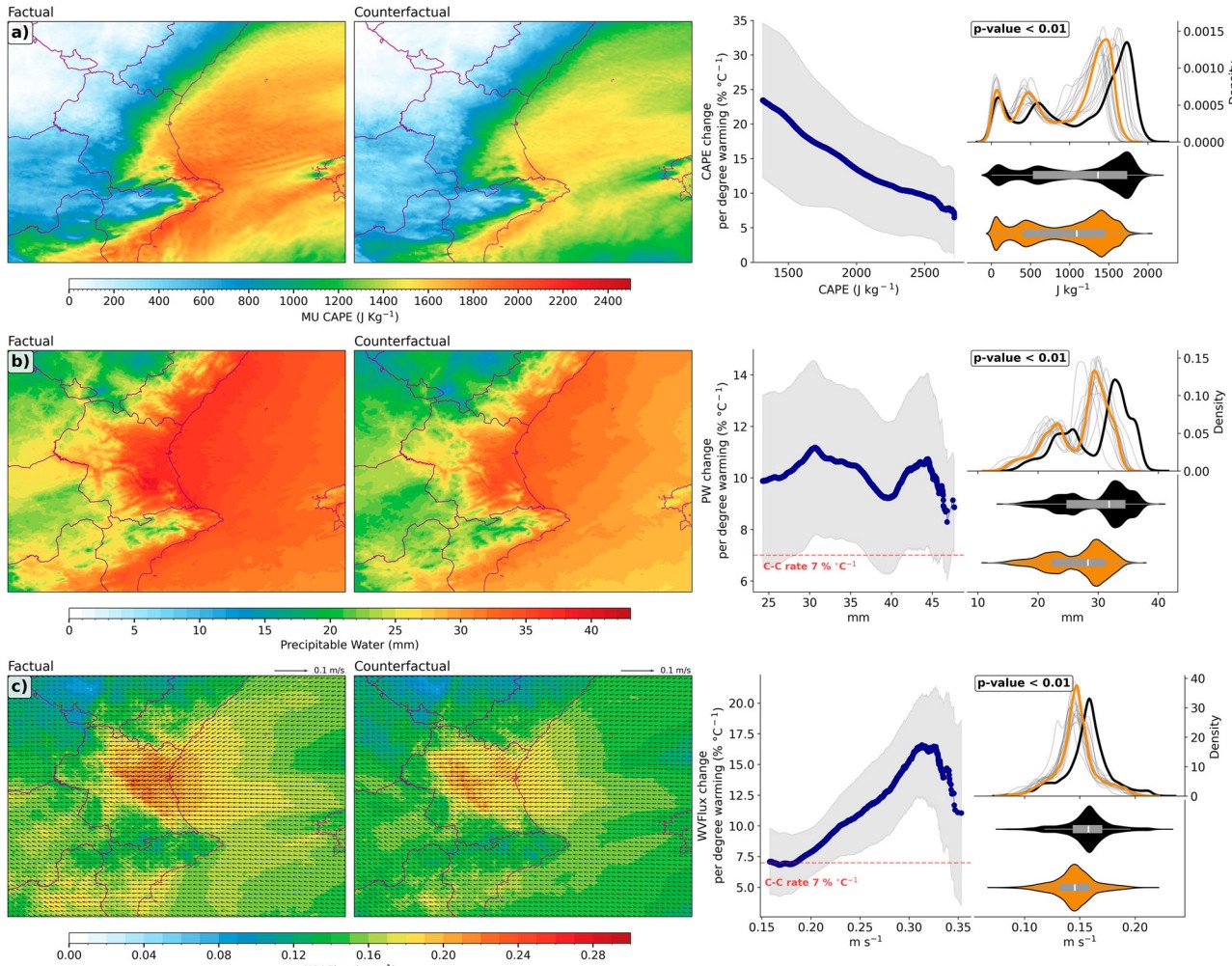

**Fig. 3 | Warming climate increases the moisture content and convective available potential energy.** Comparison between the factual and counterfactual simulations during the storm period for **a)** time average of Most Unstable Convective Available Potential Energy (MUCAPE), **b** Precipitable Water (PW) and **c** Water Vapor Flux (WVFlux) between surface and 700 hPa. Adjacent plots on the right side show the percentage change per °C of warming (the difference between the two climates states is +1.08 °C), the PDF for each simulation (black: factual simulation; orange: counterfactual ensemble mean simulation; gray: each counterfactual anthropogenic forcing simulation), and the violin plot. The legend in PDF shows the *p*-value from a Mann−Whitney U test to assess that there is a statistically significant difference between the two distributions.

increasing the intensity, spatial extent, and duration of rainfall through positive feedback mechanisms[23,49,54,69,70].

ACC is also affecting cloud microphysics (Fowler et al. 2021). The more intense convection (Fig. 4a) in the factual simulation results in ascending motions of stronger intensity and covering a wider area, indicative of robust convective processes due to the large amounts of MUCAPE and moisture content in the Valencia region (Fig. 3a). Moreover, more vigorous updrafts drive vertical moisture fluxes essential for heavy rainfall formation[71,72]. On the other hand, the pre-industrial simulations generally display less intense and more spatially dispersed updrafts, although counterfactual simulations show stronger updrafts compared to the factual simulations at certain grid points and in some climate model members (Fig. 4b). However, the increased spatial extent in the present-day climate simulation indicates that ACC amplifies convection. This larger storm footprint (i.e., strong updrafts in a wider area) in the factual climate can scale up the area exposed to high rainfall rates and boost downstream hydrological impacts (e.g., larger contribution to river and ravine catchments which promotes longer flood-prone areas). This aligns with studies over the Alpine-Mediterranean region that report changes in heavy precipitation events scale (larger events), propagation and convective

organization[39,40]. The median value of maximum updraft intensity increases by +11.9%. These significant differences not only affirm the theoretical increases and enhancement due to the warmer climate but also provide empirical evidence of ACC tangible impacts on convective storms and associated heavy rainfall events[54,70,73].

The more intense convection (Fig. 4a, b) and higher moisture content (Fig. 3b, c) in the factual world result in a higher ratio of graupel in the cloud, which in turn increased rainfall rates and solid precipitation in the Valencia event. This aligns with the projected future changes in convective storms under a warming climate[23,54,70,74–76]. The factual run shows a broad swath of large graupel concentration values stretched from southeastern to northwestern areas of the domain (Fig. 4c and Supplementary Fig. 4). This indicates an extremely vigorous convective activity capable of producing very large amounts of graupel. In contrast, the counterfactual simulation depicts a narrower region of high graupel and hail content, and reduced graupel formation affecting much of the domain. The median of maximum graupel in column increase from pre-industrial to present-day climate is +32.4%. The positive feedbacks -stronger updrafts, higher ratio of graupel and hail- promoted by ACC can contribute to more intense precipitation events, when graupel melts and

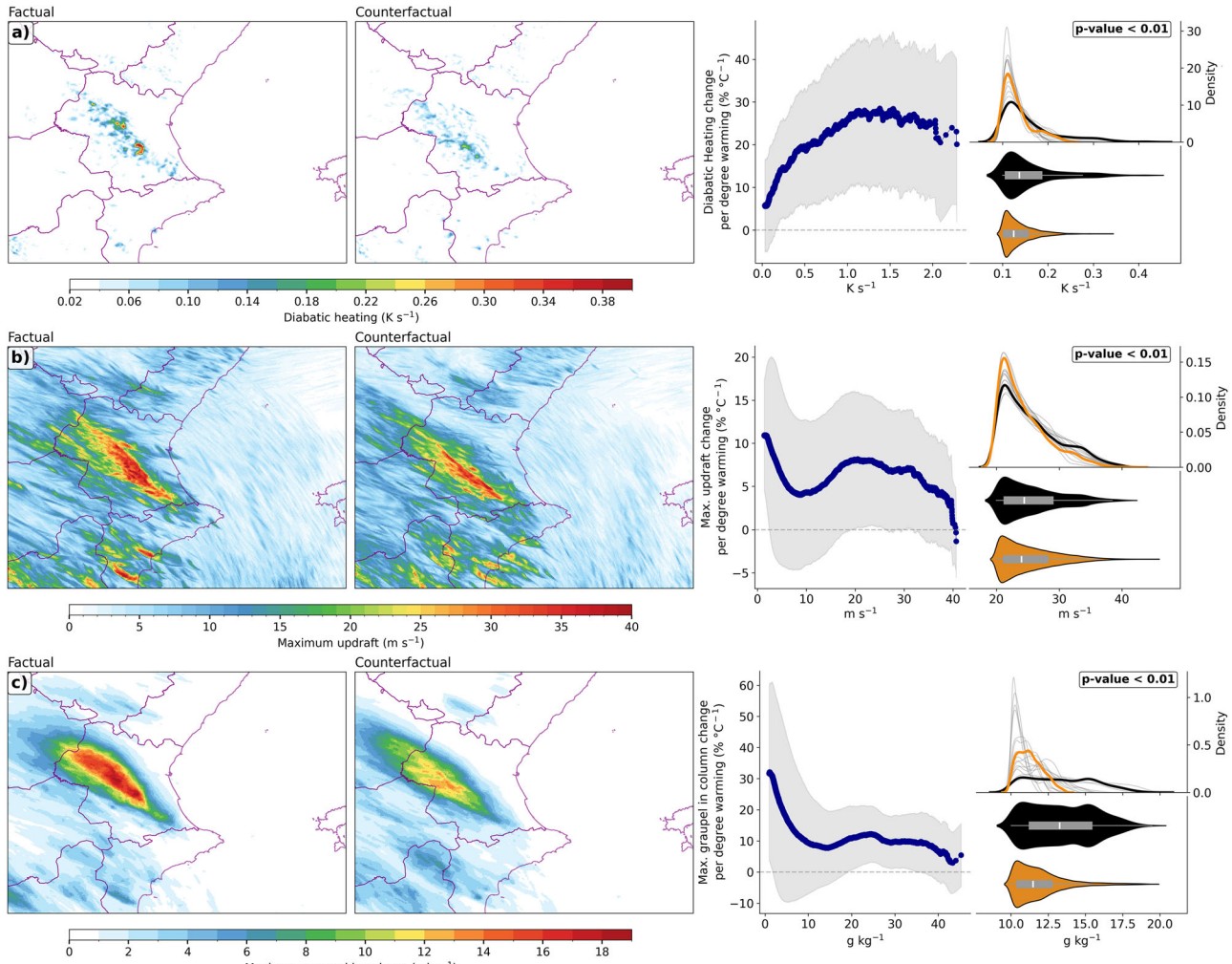

**Fig. 4 | Human-induced climate change promotes more vigorous convection and stronger updrafts.** Comparison between the factual and counterfactual simulations during the storm period for **a** time mean of the maximum diabatic heating in the vertical column, **b** the maximum updraft speed and **c** the maximum graupel concentration in the vertical column. Adjacent plots show their percentage change per °C of warming (the difference between the two climates states is +1.08 °C), the Probability Density Function for each simulation (black: factual simulation; orange: counterfactual ensemble mean simulation; gray: each counterfactual simulation), and the violin plot. The p-value from a Mann–Whitney U test indicates that there is a statistically significant difference between the two distributions.

coalesces with other hydrometeors to generate significant rainfall (Fig. 2b, d), which could lead to super-Clausius-Clapeyron scaling[77,78]. However, the link between graupel and precipitation efficiency is known to be model- and context-dependent[79,80]. Hence, the potential for super-CC should be interpreted as conditional on storm structure (Fig. 4a), MUCAPE (Fig. 3a) and precipitation efficiency[81,82] (Supplementary Fig. 7) rather than on graupel enhancement.

In addition, the elevated temperature of the warm cloud layer (i.e., the layer between the lifting condensation level and the 0 °C isotherm) in the current climate simulation, relative to the pre-industrial simulation, facilitates increased precipitation through collision and coalescence (i.e., the warm rain processes are favored) (Supplementary Fig. 6). The median of warm cloud layer height increase from pre-industrial to current climate is 9.6%. The presence of deep warm layers, which have been observed in several flash-flood episodes[54,70,78,83], produces larger droplets that descend more swiftly and are thus more likely to reach the surface, which in turn supports the expansion of regions exhibiting high rainfall rates (Fig. 2b, c).

All the changes in the cloud microphysics shown here are associated with a substantial increase in precipitation efficiency in the factual simulation (Supplementary Fig. 7). This larger precipitation efficiency in the present-day climate, by +12.6% in the

factual simulation, favored an increase in the rainfall rates in the Valencia event.

## Discussion

This attribution study highlights the substantial influence of ACC-driven moisture increases and atmospheric instability on convective storm dynamics, emphasizing their crucial role in intensifying the October 2024 extreme precipitation event in Valencia (Spain). Although the factual precipitation field is slightly displaced westward relative to the observations, the findings presented in this regional study are consistent with broader evidence that human-induced climate change is intensifying the global hydrological cycle[42,84]. While our analysis focuses on the Valencia region, the results support the idea that such localized increases in flash-flood events may be part of a wider global trend.

ACC notably increases both the intensity and the spatial extent of the extreme rainfall Valencia event. Under present-day climate conditions, 6-h rainfall rates and extreme rainfall thresholds (P90, P95, AEMET red warning for heavy rainfalls in the Valencia region and P99) show a significant spatial expansion. Figure 5 outlines conceptually the main changes found in this study, showing increments in the analyzed factors with respect to the counterfactual climate conditions. The

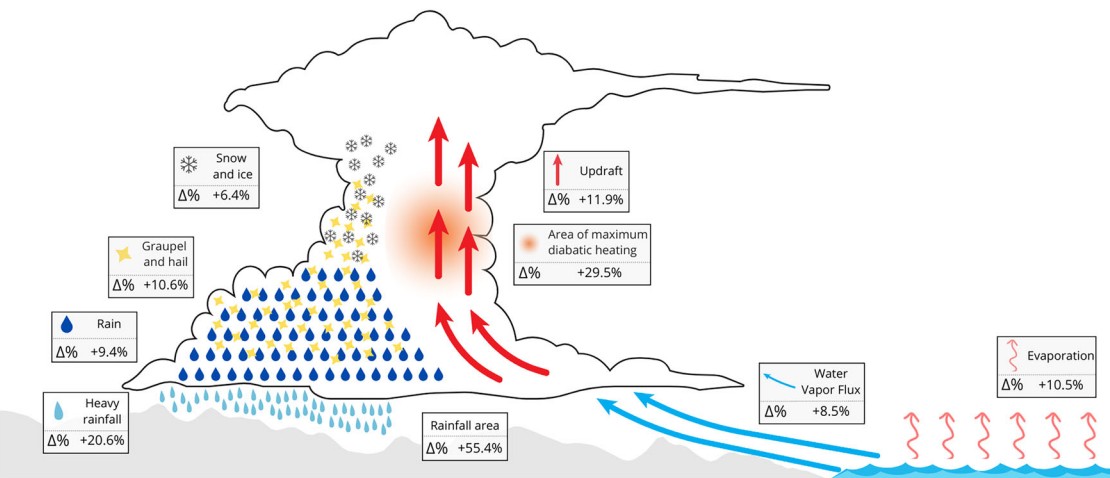

**Fig. 5 | Extreme heavy rainfall events intensify with human-induced climate change in the Valencia case study.** Heavy rainfall events intensify due to enhanced moisture content, which increases the latent heat release, and causes stronger positive vertical motions. These changes promote more intense microphysics processes, stronger heavy rainfall (6 h) and increases the total rainfall area.

enhanced moisture content, in combination with largely unchanged wind fields, leads to more efficient moisture transport and reduced the entrainment. Together, these factors lead to the intensification of this rainfall event, as highlighted by the changes in PW and WVMXR in comparison to the pre-industrial era conditions. These results are in line with the Clausius-Clapeyron relationship[29–32]. Moreover, the strengthened horizontal WVFlux (+8.5%) underscores that moisture transport enhancement is one of the primary drivers of intensified rainfall, rather than changes in wind dynamics. The increased CAPE illustrates larger atmospheric instability under current climate conditions, promoting more vigorous vertical transport of humidity, convection and stronger latent heat release. All these nonlinear processes collectively contribute to intensified rates exceeding the Clausius-Clapeyron scaling (see Limitation Statement in the Methods section). The increase in atmospheric content of water vapor significantly promotes changes in the storm dynamics, including latent heat release (+29.5%), vertical velocities (+11.9%) and cloud microphysics (-+9%). In addition, the simulated increase in the warmer cloud layer height enhances the warm rain processes. These changes result in increased precipitation efficiency and larger areas (+55.4%) experiencing intense rainfall rates (+20.6%), resulting in flash-flood risks.

Therefore, this study highlights that future projected scenarios for extreme rainfall events are already becoming evident. Such findings emphasize an immediate need to accelerate the development and implementation of climate change adaptation strategies, enhancing urban resilience in response to this growing threat, particularly within the Western Mediterranean region.

## Methods
### Datasets
Simulations from 15 CMIP6 GCMs[85] (Supplementary Table 1) are selected to provide the four meteorological variables required for applying the methodology used in this study: near-surface air temperature, air temperature, skin temperature, and specific humidity. These GCMs offer monthly data for the historical period (1850–1879 and 2009–2014) and for the SSP2-4.5 scenario (2015–2038).

For the validation of the factual simulation, hourly precipitation observations were collected from 256 weather stations within the study region (Fig. 1b). The Spanish Meteorological Agency (AEMET) provides records from 134 stations, the Valencian Meteorological Association (AVAMET) provides 107 stations, and the private company SISRITEL operates 15 stations. All raw meteorological datasets used in this study underwent rigorous quality control checks to ensure accuracy and consistency[86]. Additionally, ERA5 reanalysis data[87] is used to analyze the synoptic pattern conditions of 500 hPa geopotential height and mean sea level pressure on the selected day.

The Weather Research and Forecasting (WRF) model v.4.5[88] is used to simulate the heavy precipitation event in current climate (hereafter, factual simulation) and pre-industrial climate (hereafter, counterfactual simulation). The simulations included two nested domains over the Valencia region with horizontal grid spacing of 3 km and 1 km, and 65 hybrid vertical levels; the latter better resolves deep-moist convection and improves precipitation resolution. Several different configuration and initialization time tests have been carried out to obtain the most realistic factual simulation (the Supplementary Material includes a comparison with different factual and counterfactual initializations). All initialization times exhibit similar relative changes between factual and counterfactual runs, which provides confidence that the main conclusions are not sensitive to the specific initialization chosen. Finally, the main WRF model settings are: the WRF single-moment six-class scheme for microphysics[89], the Yonsei University scheme for the planetary boundary layer[90], the Dudhia short-wave scheme[91] and the rapid radiative transfer model long-wave scheme[92] for radiation. Cumulus clouds are explicitly computed by the model in both domains. The factual run is driven by initial and lateral boundary conditions from the ECMWF-IFS analysis with a 0.1° horizontal resolution, 27 vertical levels and 6-h temporal resolution. The simulations run for a 36-h period, setting a 12-h spin-up and an initialization time set at 18:00 UTC on 28 October 2024. The factual simulation is shown to capture adequately the mesoscale convective systems and the rainfall pattern in the Valencia region as reflected by the observational data (Fig. 1b).

### Storyline approach by Pseudo-Global Warming Approach
The storyline approach by the Pseudo-Global Warming (PGW) experiment is a method widely adopted in previous studies of regional climate change (e.g., refs. 22,26,93). This method adds a climate perturbation signal to the baseline conditions for the period of interest[94–96]. Herein, this approach is applied to analyze the ACC signal contribution to the extraordinary rainfall associated with the mesoscale convective system studied here, considering the historical and SSP2–4.5 scenarios[97] from 15 CMIP6 climate models (Supplementary

Table 1). Several prognostic variables from the ECMWF-IFS analysis, used as initial and boundary conditions are modified; all of them are purely thermodynamic: air temperature, specific humidity, 2-meter temperature, and skin temperature. Moreover, the greenhouse gases are perturbed to reflect preindustrial conditions modifying the GHG concentration in the WRF source code[22]. First, for each GCM, anthropogenic forcings (Δ in Eq. (1)) have been computed as the difference between two long-term monthly mean climate periods: 1850–1879 and 2009–2038. To obtain a climate timeframe where the extraordinary rainfall event is centered into a 30-year climatic period, we concatenated the CMIP6 historical experiment from 2009 to 2014 with the intermediate pathway scenario (SSP2–4.5) from 2015 to 2038:

$$\Delta X_m = \overline{X_{(1850-1879),\,m}} - \overline{X_{(2009-2038),\,m}} \qquad (1)$$

where $m$ is the month, and $X$ denotes one of the prognostic variables. Besides the climate forcing for each model, the ensemble mean, from the different CMIP6 climate models used herein, is computed and analyzed as one more climate model.

Finally, the initial and boundary conditions for the storyline simulations are then generated by adding the mean monthly climate forcings to the 6-hourly IFS data to generate the pre-industrial-like (counterfactual) climate initial/boundary conditions (Eq. (2)).

$$X^{PGW} = X^{IFS} + \Delta X_m \qquad (2)$$

where $X^{IFS}$ are the IFS initial and lateral boundary conditions and $X_m$ is the ACC signal forcing obtained in Eq. (1). This is computed for each climate model used in this study (Supplementary Table 1). The storylines conducted by the modified initial and lateral boundary conditions are compared against factual simulation to quantify how the ACC enhances the heavy rainfall event. Compared with previous research studies that used PGW simulations, this survey employs climatic forcings from 15 individual climate models and generates a multimodel ensemble mean to simulate the counterfactual-like event, rather than relying solely on the ensemble mean. This approach allows us to assess how each climate model attributes the extraordinary rainfall event to ACC, while also providing a larger dataset for more robust statistical analysis and capturing the uncertainty related to ensemble mean simulation.

## Assessment metrics

This study analyzes (i) rainfall intensity and area, (ii) changes in the moisture content and fluxes and (iii) changes in physical mechanisms that control heavy rainfall during the storm period in the simulation (13 h, October 29th 04-16 UTC) within a domain centered in the Valencia region (see Figs. 2–4).

The rainfall is analyzed deriving the total precipitation from the differences between WRF model 1-h outputs (1-h rainfall intensity) and aggregating them into 6-h rainfall intensities. The rainfall area is derived from total precipitation factual simulation by establishing four objective-statistical thresholds: 90th percentile rank (P90), 95th (P95), 180 mm and 99th (P99). The 180 mm threshold is set because it corresponds to the red warning from the AEMET for heavy precipitations in Valencia's region[98]. To evaluate the ACC influence, the rate of increase from counterfactual (in each climate model simulation) to factual simulation is computed by the Eq. (3):

$$\%_{increase\,m} = \frac{Factual_m - Counterfactual_m}{Counteractual_m} \cdot 100 \qquad (3)$$

where $m$ represents each WRF model simulation with its corresponding $m$-$th$ climate model forcing.

Several diagnostic variables and postprocessed parameters are considered to study and analyze their relationship to heavy rainfall:

water vapor mixing ratio, rain water mixing ratio, graupel mixing ratio, ice and snow mixing ratio, MUCAPE, PW and WVFlux. All the mixing ratio parameters are a direct output of the model. CAPE and PW are computed using the diagnostics computations in the python library *wrf-python* (wrf.cape_2d and wrf.pw, respectively)[99].

WVFlux is the flux of water vapor at each model level. This parameter is used, for example, to evaluate the vertical profile of water vapor in atmospheric rivers and extreme rainfall events[100,101]. The WVFlux (in units of m s⁻¹) is computed by the Eq. (4):

$$WVFlux = \sqrt{(u \cdot q)^2 + (v \cdot q)^2} \qquad (4)$$

where $u$ and $v$ are the wind components (m s⁻¹), $q$ is specific humidity (kg kg⁻¹) at every model level. To evaluate the changes in the physical mechanisms, we use multiple WRF postprocessed diagnostics which controls heavy rainfalls events: maximum positive vertical velocity (updraft), maximum graupel and hail concentration in the column, microphysics latent or diabatic heating, maximum updraft helicity and precipitation efficiency (PE). PE is defined as the ratio of the surface precipitation rate to the total vertically integrated condensation rate, including liquid and ice (P/C, kg m⁻² s⁻¹), yielding a dimensionless efficiency[102]. To compute PE, we follow the first approximation methodology of[102], in which condensation is derived from state variables. The rest of the variables are directly produced by the model through the activation of the National Severe Storm Laboratory (NSSL) WRF diagnostics[103].

The Clausius-Clapeyron scaling analysis is carried out to determine how high is the extreme precipitation (and rest of the parameters) response to the temperature change between factual and counterfactual simulations. First, we calculate the temperature difference before the event started in Valencia between the two climates, resulting in an average increase of +1.08 °C. Then, we sort the hourly precipitation (and rest of the parameters) in the storm period and calculate the difference between the two climate states (Eq. (3)). Finally, we divide such %increase by the temperature change, which results as % change per degree warming.

The non-parametric Mann–Whitney U test[104] is used to establish differences (99% confidence) between the factual and counterfactual simulations for the variables abovementioned. All differences described as significant in the manuscript are tested and confirmed at the $p < 0.01$ level.

## Limitations

This work has some limitations regarding the modeling setup, PGWA constraints and the way we evaluated the change per °C of warming. First, PGWA constrains the large-scale circulation to that of the observed event and primarily imposes thermodynamic changes. Therefore, it cannot explore alternative cut-off low tracks or dynamical regime changes that a colder (attribution approach) or warmer (future changes) climate might favor. Our result should be interpreted as conditional ("*if an event like the Valencia floods had occurred under a pre-industrial climate, how would it have changed?*") rather than probabilistic statements about the probability of happening[105–107]. Second, focus only on a single high-impact episode limited the representativeness, because internal variability and seasonal dependence are not fully introduced[106]. Third, model resolution and physics configuration introduce numerical uncertainty. Despite convective-permitting grid spacing (1-km), complex orography and sea-breeze interactions along the Valencian coast can still be misrepresented and impact the way the precipitation extremes are resolved, which are also sensitive to microphysics and boundary layer schemes[108,109]. Since this is a mesoscale phenomenon with inherently limited predictability, it is recommended to use an ensemble framework incorporating stochastic perturbations to the initial conditions or to the model itself[110,111]. Therefore, an ensemble prediction system to capture the

meteorological uncertainty, in addition to the climatic uncertainty addressed in this study, would be more appropriate. Although we used 15 CMIP6 GCMs to assess the uncertainty related to climate models, an additional uncertainty source relies on the way PGW forcings are constructed, which could have an impact on moisture availability and mesoscale triggers[96,107]. Finally, the "change per degree warming" label is used here as a concise description of the relative difference between two discrete physical states (*present-day* vs. *pre-industrial–like* conditions). It does not imply linearity beyond this range. Extrapolation to a wider range of warming should be made with caution since short-duration extremes can depend on temperature, storm type and dynamics.

## Data availability

The following GCM data sets used in this study are available through the CMIP6 repository (https://esgf-node.llnl.gov/projects/cmip6/). ERA5 reanalysis is available from the Copernicus Climate Change Service Climate Data Store[87] (https://doi.org/10.24381/cds.bd0915c6). Meteosat Second Generation data is available from the EUMETSAT Store (https://data.eumetsat.int/) and the EUMDAC Python package (https://gitlab.eumetsat.int/eumetlab/data-services/eumdac). Shapefile data used in this work is available from the GADM data (https://gadm.org/download_country.html). PGW increments were performed using PGWERA5 v1.1[96], which can be downloaded from a Github repository (https://github.com/Potopoles/PGW4ERA5.git). Numerical simulations were performed using WRFV4.5, which can be downloaded from the UCAR Github repository (https://github.com/wrf-model/WRF.git). WRF namelists used to generate all the simulations are available in https://github.com/ccalvosa/dana_vlc_attribution. The analysis and visualization scripts will be available in a Github repository upon a reasonable request to the corresponding author.

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

## Acknowledgements

This research has been supported by the grant PID2023-146344OB-I00 (CONSCIENCE) funded by MICIU/AEI/10.13039/501100011033 and by "ERDF A way of making Europe", and the ECMWF Special Projects (SPESMART and SPESVALE). This work is supported by the Interdisciplinary Mathematics Institute of the Complutense University of Madrid. C.C.-S. acknowledged the grant supported by the Spanish Ministerio de Ciencia, Innovación y Universidades (PRE2020-092343). A.H.-M. is grateful for his MCI/AEI predoctoral contract (FPU18/00824). M.M.M. acknowledges financial support from Next Generation EU, Mission 4, Component 1, CUP B53D23007360006, project "WIND RISK".

C.A-M. acknowledges support from the GVA. PROMETEO Grant CIPROM/2023/38; CSIC-LINCGLOBAL Ref. LINCG24042; and CSIC's PTI-Clima. We would like to thank Dr. Linda van Garderen and two anonymous reviewers for their valuable comments to improve this work and for the effort they made to review this manuscript.

## Author contributions

C.C.S. conceptualized and conducted the research, performed the analysis and drafted the article. J.D.F. conducted the research, performed the analysis and drafted the article. J.J.G.A. contributed to the research, interpretation of the results and reviewed the article. A.H.M. contributed to the interpretation of the results and drafted the article. M.M.M. contributed to the research, interpretation of the results and reviewed the article. C.A.M. contributed to the research, interpretation of the results and reviewed the article. A.F.P. contributed to the research, interpretation of the results and reviewed the article. A.M.M. contributed to the research, interpretation of the results and reviewed the article. P.B. reviewed the article and contributed to the research. A.M. contributed to the research, interpretation of the results and reviewed the article. M.L.M. contributed to the research, supervised the work, reviewed the article and contributed to the interpretation of the results.

## Competing interests

The authors declare no competing interests.

## Additional information

[1]Department of Applied Mathematics. Faculty of Computer Engineering, Universidad de Valladolid, Valladolid, Spain. [2]Centro de Investigaciones sobre Desertificación, Consejo Superior de Investigaciones Científicas (CIDE, CSIC-UV-GVA), Climate, Atmosphere and Ocean Laboratory (Climatoc-Lab), Moncada, Valencia, Spain. [3]Spanish State Meteorological Agency (AEMET), Department of Science, Madrid, Spain. [4]Instituto Pirenaico de Ecología, Consejo Superior de Investigaciones Científicas (IPE-CSIC), Zaragoza, Spain. [5]Laboratorio de Climatología y Servicios Climáticos (LCSC), CSIC-Universidad de Zaragoza, Zaragoza, Spain. [6]National Research Council of Italy, Institute of Atmospheric Sciences and Climate (CNR-ISAC), Padua, Italy. [7]Institute of Atmospheric and Climate Science, ETH Zurich, Zurich, Switzerland. [8]Consejo Superior de Investigaciones Científicas (CSIC). Instituto de Geociencias (IGEO), Madrid, España. [9]Department of Earth Physics and Astrophysics, Faculty of Physics, Complutense University of Madrid, Madrid, Spain. [10]Interdisciplinary Mathematics Institute. Universidad Complutense de Madrid, Madrid, Spain. ✉e-mail: carlos.calvo.sancho@uva.es

