## [Transparent Peer Review file · Nature Communications]

Human-induced climate change amplification on storm dynamics in Valencia's 2024 catastrophic flash flood

Corresponding Author: Dr Carlos Calvo-Sancho

Version 0:

Reviewer comments:

Reviewer #1

(Remarks to the Author)

Review for: Climate change unleashed: physical-based attribution analysis proves human-induced amplification of Valencia's deadly flooding

Calvo-Sancho et al.

This study attributes the influence of climate change on the extreme rainfall event that occurred near Valencia on 29 October 2024. The event was truly remarkable and caused substantial societal impact. The study is timely and, whilst other groups have made similar attempts to attribute the influence of climate change on this event, the novelty of this work is in probing the convective scale dynamics of the storm and the role of microphysical processes.

The manuscript is concise and well written (with the exception of some missing detail described below), and the figures are clear and generally appropriate. My only methodological concern relates to the robustness of the results as I describe below. If the authors are able to address this concern, along with my more minor comments on the presentation, I would be very happy to see the work published.

Major comment

- The authors compare one simulation of the Valencia rainfall event (the "factual" simulation) with an ensemble of pseudo-global warming simulations (the "counterfactuals") produced by modifying the initial and boundary conditions with the aim of removing the influence of the warming experienced since preindustrial times. The ensemble of counterfactuals is generated by taking changes in temperature and humidity between pre-industrial and present day periods from a set of CMIP6 simulations. As such, the method aims to sample uncertainty associated with the amount of climate change experienced between pre-industrial times and the present day. This method will also, perhaps unintentionally, act to sample some variability associated with the chaotic evolution of the atmosphere in the same way that an ensemble NWP forecast does. This is useful as it adds to the robustness of the results by sampling a range of possible outcomes. My concern, however, is that only one factual simulation is considered, and furthermore, that this factual simulation has apparently been hand-picked from several simulations because it best reproduces the extreme high rainfall anomalies observed (see Methods: "Several different configuration tests have been carried out to obtain the most realistic factual simulation."). Given this, I am left unsure how robust the results are to the choice of factual simulation used. Can the authors provide reassurance that the results are robust in an ensemble forecasting sense? One way to do this could be to run a set of simulations starting at different initialisation times in the run up to the event. Do they all exhibit the same changes relative to the counterfactuals?

Minor comments

- The figure captions lack important details. Please review these to ensure they fully describe the figures, paying particular attention to:

o Figure 1 caption: What is the accumulation period for the rainfall map?

o Figure 2 caption: Panel a: Are these distributions a single time or several (if so, what time(s)?). What does 'theoretical quantile' mean? Panel b: Please describe what is shown here (I think I understand, but only after reading right to the end of the Methods section – it is not clear from the caption). Panel c: The meaning of this sentence is not clear, how about

“Percentage increase in the total area exceeding thresholds equal to the P90, P95 and P99 percentiles from the factual simulation and the AEMET red warning in the Valencia region.” Panel d: Please emphasise for clarity that the factual panel is a single simulation but the counterfactual is the mean of the ensemble of simulations. What is the p-value showing here?
o Figure 3 caption: Do you mean convective available potential energy? What time is shown in all of the panels? Are the PDFs constructed from a single time? What does “non-climate CMIP6 model forcing” mean? You say the legend “shows statistical significance”, but of what? Please spell this out, e.g. “shows the p-value from a Mann-Whitney U test which aims to assess whether there is a significant difference between the two distributions”.

o Figure 4 caption: “Human-induces” -> “Human-induced”. Panel a: What is maximum diabatic heating? Is this the maximum in the vertical, or in time. If in time, what level is shown? If in the vertical, what time is shown? Panel b: “Maximum updraft” -> “Maximum updraft speed”. What does “non-climate CMIP6 model forcing” mean? All three PDFs look to only show values above a certain threshold, please specify what this is. I find it very hard to interpret what the p-values mean in this case because you are asking if the shape of the tail above the threshold is different between the different simulations. Would you expect it to be? Surely a more obvious question is how many more data points lie above the threshold?

- The description of the simulations lacks some important details:

o What domains were used for the 3km and 1km nesting?

o What time were the runs initialised and how long were they?

o Has the performance of this version of the model at simulating extreme convective events been evaluated? Please provide relevant literature.

- An important caveat to this work is that all the results presented are conditioned on the same large scale dynamical setup occurring in different climates. If the frequency of occurrence of such setups changes (or their dynamical intensity changes) then these will have important impacts which are not considered here. Whilst this is well known within the attribution community, I would strongly encourage the authors to be explicit about this point for clarity. Can they draw from the literature or IPCC assessment reports to comment on the state of understanding and uncertainty in this respect for these types of events? At the very least, the caveat should be mentioned in the abstract and the conclusions.

Typos / points of clarification

Abstract: Several acronyms are used in the abstract without being introduced. Please define these.

L43: “in Turis official weather station” is not good English. How about “in the official weather station at Turis”? Also, over what period was the 771.8mm accumulation?

L43; “and record-breaking in one hour in Spain” does not make sense. How about “and breaking the record for one hour rainfall accumulation in Spain”?

L45: It is worth saying the WRF model is run at km-scale resolution.

L46: “sourced with CMIP6 models” is unusual terminology. Please either expand to say in what way CMIP6 models are used here, or (probably better) just remove the phrase and leave the details to the main text.

L48: “Observed climate change led to...” is not accurate. You explicitly use simulated changes in climate to drive the model. How about the more precise ‘Compared to pre-industrial conditions, present-day conditions lead to...’.

L49: In what way is 20%/C consistent with Clausius-Clapeyron? Please expand.

L51: You say “stronger vertical velocities” but do you also mean larger CAPE and stronger diabatic heating?

L64: Presumably breaking Rossby waves are only relevant for extreme precipitation events in the extra-tropics?

L66: “de Vrier” -> “de Vries”

L96: Missing word near “atmospheric river-like from”?

L110: “using the physical-based attribution approach” Please add a sentence or to outline the main differences between the factual and counter-factual simulations, and refer the reader to the methods section for full details.

L132: Are the areas smaller for any one member? You use a log scale so negative values wouldn't show up if they were.

L211: This whole paragraph is a repeat of the previous one.

L226: “of diabatic heating more intensity” does not make sense.

L227: “The median diabatic heating increase from pre-industrial to present-day climate is 29.5%”. What does this mean? Is this out of all grid points with non-zero diabatic heating?

L317: What does “precipitation numerical resolution” mean?

L338: “we had to concatenate” -> “we concatenated”.

L347: As understand it, there is only one factual simulation, but you say “are compared against factual simulations”. Please clarify.

L386: What is “the storm period”?

Extended Data Figure captions 3-7: Shouldn't these say “As in Extended Data Figure 2”?

Reviewer #2

(Remarks to the Author)

See the attached report

Reviewer #3

(Remarks to the Author)

My full review is uploaded in pdf form.

Version 1:

Reviewer comments:

Reviewer #1

(Remarks to the Author)

Second review for: Human-induced climate change amplification 1 on storm dynamics in Valencia's 2024 catastrophic flash flood (amended title)

Calvo-Sancho et al.

I'd like to thank the authors for responding comprehensively to my reviewer comments, and in particular for providing a comparison against simulations with a range of lead times in order to provide confidence in the robustness of the results. I have read those parts of the revised manuscript relevant to my comments and am happy with the changes made.

I just have one very minor comment which the authors may wish to consider:

L354: Please briefly explain what you infer from the set of simulations with different initialization times (e.g., as you describe in your response doc). Also, I think 'initialization times' would be clearer terminology than 'initializations' (also L353).

Reviewer #2

(Remarks to the Author)

The attached report includes few more comments for the authors and a recommendation for the Editor for publication.

Reviewer #3

(Remarks to the Author)

I have placed my comments in appendix.

Version 2:

Reviewer comments:

Reviewer #3

(Remarks to the Author)

Dear Authors,

Thank you for your final corrections and for the clear explanations addressing my comments from the second review. I am satisfied with the revisions and consider the manuscript ready for publication.

Congratulations on this work.

Kind regards,

Linda van Garderen

NCOMMS-25-38404

“Climate change unleashed: physical-based attribution analysis proves human-induced amplification of Valencia's deadly flooding” by Calvo-Sancho et al.

The authors attempt to frame the Valencia extreme event occurred in October 2024 in the context of impact of global warming. They use the model WRF to simulate the event, presenting a very quick assessment of the precipitation field against the rain gauges.

They then perform a comparison between the simulated event and a “synthetic” preindustrial scenario (built through CMIP6 models climate forcing with a Pseudo-Global Warming approach), quantifying the impact of the global warming on the intensity of the catastrophic event, through the study of the change of different storm drivers, and so claimed as “physical-based” attribution through the approach of the storylines.

Although the method is already known, the 2024 Valencia extreme event considered is still poorly studied, as well as its attribution to global warming. Some interesting aspect arises, in terms of climate change impact, considering the sub-daily scale and driver variables unusual in attribution studies; apart these, the novelties the author intend to share with the scientific community for advancing knowledge are hard to be captured and should be better highlighted. The comparison with existing literature should be enriched and the Method section should be improved (details on simulation configuration should be added or model namelists should be available to make the experiments reproducible). Some results should be discussed more in depth than simply reporting the plot evidences. Some of the cited papers are missing in the bibliography and a revision of English is needed.

Below some specific comments which I think can help the authors to improve the paper, which can be considered for publication after major revisions.

Comments

L76-79 I would say that also the method of Faranda et al., 2024 is physical-based as it is based on different hazard variables over a statistical base (also weighting the role of low-frequency modes of natural variability); they also reported about your same event. Your method is related to the single event analyzed (through well known PGW technique). I would better highlight the novel aspects you are presenting here, as well also discuss those ones that can eventually be generalized in the framework of climate change impact on wet hazard. Apart the specific event, for example I have appreciated that you have considered also the sub-daily scale, which is not as usual in attribution studies, but it has been suggested to be intensifying due to warming more sensibly than longer time-scale extremes (<https://doi.org/10.1029/2020GL089300>)

L110 and L123 (and following) The authors introduce the “counterfactual” event, i.e. the event in a pre-industrial climate, then using it for comparison with the factual extreme in the following sections. I suggest a concise mention on how it is built in the “Introduction” or in the “Results” paragraph, then referring the following “Methods” section for a more in depth explanation.

Figure 1 It would be useful to add in panel “b” locations of sensors in panel “c” (Turís and Poio ravine); panel a: what is the data source? Moreover, the evaluation analysis against observed precipitation is very quick and based only on total precipitation map. The reader should trust that the event is overall fairly reproduced based on this plot (1b). The discussion here can be improved, even not necessarily adding further graphics but expanding the evaluation over other aspects about the storm reproduction against observations, which is not clear if has been performed (and not shown) or it is strictly limited on what shown on Figure 1.

Section “Changes in rainfall intensity and area”: It would be good to have also a comparison (at least as discussion) between the two events large scale conditions.

Figure 2, Some of the comparison criteria are explained in the figure captions but not in the main text L132-140 (for ex. “The total rainfall percentiles are calculated from the factual simulation”). Moreover some further discussion about how much your results here are in line with existing literature or not would help to better highlight any novel aspect arising (for ex., but not only, <https://doi.org/10.1016/j.wace.2025.100781>, <https://doi.org/10.1029/2024GL112392>, <https://doi.org/10.21203/rs.3.rs-2065981/v1>, doi.org/10.1007/s00382-023-06901-9, <https://doi.org/10.1029/2023GL105143>)

L132, L286 confirming existing literature, that should be cited (ex. but not limited to: <https://doi.org/10.1038/nclimate2258>, <https://doi.org/10.1175/JCLI-D-16-0808.1>). I suggest to improve the discussion about these results.

L140 Any comment about the GCM giving larger extremes in the counterfactual event?

L183 Figure 3: what is the vertical portion of the air column considered for PW and WVflux?

L190 Extended data Figure 2: what is the level shown?

L198-200 The link between the two statements and speculations, should be discussed more in depth.

L201 and L204 zonal flux; is WVFlux integrated?

L99, L206-207, L281, L299, L371, L388 citations missing in the references list.

L226 (and following), L329, L330 citations missing in the references list or typo in referencing.

L211-220 Copy of the previous paragraph.

L238-39 You can speculate a bit here linking to a possible spatial increase of the risk/severity related to the increased storm area as well as downstream. Still in line with other literature about heavy storm sensitivity to global warming (ex. doi.org/10.1007/s00382-023-06901-9, <https://doi.org/10.1029/2023GL105143>).

Extended Data Figure 3 and 4: Check the captions. Moreover, is that one single level?

Extended Data Figure 7: are labels of violin plot x-axis right?

L282-84: From results I have understood that the increase of moisture is mainly contributing to the enhancement of its transport (due to almost unchanged winds), acting as primary driver of rainfall intensity. If I am correct, maybe this sentence needs some rephrasing.

Figure 5 caption: clarify “higher microphysics” . Changes in the figure are related to the Valencia case study and this should be specified.

L335 Please add details on how the GHG perturbation is built (or document from literature), so that your experiment is replicable.

Minor comments

L60 The very first sentence of the introduction may sound misleading, although it is elaborated in the following lines. You may try to rephrase.

L87 The term “overwhelming” is inadequate referred to natural hazards. Better using “devastating”

L88 causing or triggering better than “promoting”

L140 “Extended Data section”

L141 please cite properly. The DOI is not specified

L193-95 Review the English carefully

L196 “a larger extent”: do you mean spatially?

L227 more intense

L235-6 “dispersed” I guess you mean spatially; I suggest to link with following sentence (“, although” or similar) so that you can better highlight that this happens over fewer grid-points and ensemble members.

L247 “elevated” do you mean large values?

L251 affecting

L261 produces

L327 I would not use PGW acronym before its extended form in the section title.

L348 is “surveys” here used as “research studies”?

L385 only hourly precipitation?

Review: 'Climate change unleashed: physical-based attribution analysis proves human-induced amplification of Valencia's deadly flooding'

I would like to thank and congratulate the authors for their hard work and for contributing to the growing body of knowledge in extreme event attribution science. Out of appreciation for the effort invested in this study, I have reviewed the manuscript with care. The analysis shows clear potential, and with revisions to improve clarity and completeness, it could make a valuable addition to the field.

Goal and Summary of Article

The primary objective of this paper is to attribute the role of anthropogenic climate change in the meteorological conditions that led to the Valencia flash flood of October 2024. To achieve this, the authors employ a storyline-based attribution approach using a Pseudo-Global Warming (PGW) framework within a regional climate model, driven by ECMWF-IFS reanalysis data. The counterfactual (pre-industrial) simulations are constructed by modifying the boundary conditions with multi-model mean changes derived from CMIP6 GCM data.

The attribution analysis focuses on changes in precipitation intensity and spatial extent, as well as shifts in key atmospheric moisture variables, including CAPE, precipitable water, and water vapor flux. Additionally, the study examines physical mechanisms such as maximum updraft and graupel formation. Many of these variables are assessed in relation to the Clausius-Clapeyron scaling rate of approximately 7% increase in moisture per degree Celsius of warming.

Title

I'd suggest rethinking the title, as the phrase 'Climate change unleashed' feels a bit sensational for a scientific paper. Since this work is aimed at peers, a more neutral tone would likely strengthen its credibility. You could consider starting with 'Valencia's deadly flash flood' to immediately highlight the event and its relevance to attribution, without losing impact.

Abstract

The abstract appropriately sets expectations about the paper's content. However, some rephrasing could enhance clarity and better emphasize the novelty of the results. Currently, the consistency with Clausius-Clapeyron scaling is highlighted as a key finding, but this alone may not fully engage readers or convey the paper's unique contributions. It is worth highlighting in the abstract that the study employs a storyline analysis using the PGW method, as this clarification helps address potential questions from readers early on.

L44-45 The sentence referring to the event as "the major weather event" may be perceived as insensitive given the fatalities involved. Consider revising to something like: "... resulting in fatalities and ranking as the most destructive event for ..."

L49-52 Reword this section to more clearly showcase the novel aspects of your findings, rather than framing them in a way that might imply a lack of new insight. Also, since the paper focuses specifically on updraft rather than vertical velocity (despite their overlap), it would be better to consistently use the term "updraft" to align with the paper's terminology.

L54 The term “risk” was not analyzed in the study; please adjust the wording accordingly to avoid any misinterpretation.

Introduction

While the introduction reads well and provides a good overview of the event under study, several revisions are necessary to strengthen the manuscript.

The literature review is notably lacking in coverage of storyline approaches and the attribution of extreme precipitation events. Additionally, several key references are missing, which also applies to the methodology section. For instance, the manuscript discusses storylines and atmospheric uncertainty but does not cite foundational works by T. Shepherd (Shepherd, 2014, 2016). Similarly, although the study employs the Pseudo-Global Warming (PGW) approach, the original developer C. Schär (Schär et al., 1996) is not referenced. Furthermore, previous storyline analyses focused on precipitation in the Mediterranean region (Zappa and Shepherd, 2017) and flash floods (Dare et al., 2019; Orth et al., 2021) are omitted. Expanding the literature review to include these key contributions would provide readers with better context and emphasize the significance of the current research.

The detailed description of the event is appreciated and essential for establishing a baseline understanding. However, the manuscript lacks any presentation of temperature profiles. Since precipitation characteristics, CAPE, and Clausius-Clapeyron scaling are all strongly dependent on temperature, it is crucial to include temperature information. This could be visualized in various ways, but the observed temperatures, counterfactual simulations, and the differences between factual and counterfactual should be clearly illustrated both spatially and temporally—and potentially vertically as well—to fully support the analysis.

below some smaller details to look at:

L68 The term “triggering mechanisms” is introduced but never revisited or explained. Please clarify what you mean by this or consider rephrasing to avoid ambiguity.

L69 Are the results best described as “contrasting” or “uncertain”? If the uncertainty exceeds the signal, this might create a contrasting appearance. Consider adding a sentence to clarify this distinction.

L78 Please add “of” between “because” and “ACC” for grammatical correctness.

L79 Remove the phrase “In contrast,” as it is unnecessary here.

L83-84 Reword this passage to clearly articulate the specific research gap your study addresses, avoiding vague or general recommendations.

L86 Replace the word “accumulations” with “accumulative values” for precision.

L94 The current sentence is unclear—how exactly do quasi-stationary convective systems get “triggered”? Please clarify what action or change you are attributing to the trigger.

L95-97 The grammar here is awkward, making the sentence difficult to understand. Please revise.

L105-108 Demonstrating consistency with Clausius-Clapeyron scaling could also indicate that an atmospheric river is involved. Have you investigated this possibility? Can you confirm or rule it out based on your data?

L109-112 Please explicitly state that your analysis uses a storyline approach with a regional model under PGW settings. This will help readers better understand the results without needing to refer back to the methods section.

Figure 1 Suggestion: Consider replacing the color scheme in panel (a) with temperature to enhance clarity. Additionally, the unit used in panel (c) (mm/10min) may be unfamiliar—if it is not commonly used, please convert it to a more standard unit.

Figure 1b The panel currently does not effectively show the difference between observed and modeled data. Please revise the visualization to more clearly and intuitively display the comparison.

Methods

The Methods section would benefit from reorganization to improve clarity and reader comprehension. Beginning with an overview of the storyline approach would provide essential context before delving into specific details such as datasets. Presenting a general introduction to the methodology first, followed by the technical specifics, would help readers grasp the overall framework and purpose of each component. As noted earlier, some important references are missing and should be included.

Additionally, there are several questions and ambiguities that arise from the current description of the methods, which are outlined below:

1. Could you elaborate on the selection process for the CMIP6 GCMs used in your study? Additionally, to what extent might your choice of models influence the results?
2. Please specify the model domain more clearly, ideally including a map that illustrates the grid layout.
3. Clarify explicitly that the ECMWF-IFS data used are reanalysis products.
4. Explain how atmospheric dynamics are handled in the model. For example, is the upper atmosphere nudged or otherwise constrained? If so, how might this affect analyses related to updrafts?
5. Indicate whether your analysis represents averages over the entire model domain and whether this approach is consistent across all variables.
6. Many variables are expressed as percentage change per degree of warming, which provides valuable insight. However, given that your study compares only two states differing by approximately 1.08 °C, can you confidently generalize this as “per degree warming”? Without scenarios spanning a wider range of warming, the relationship beyond this specific temperature difference remains uncertain. While comparing to Clausius-Clapeyron scaling is valid for some variables, labeling changes as “per degree warming” might be misleading, as it implies a linear relationship irrespective of the warming magnitude. Strictly speaking, you are presenting percentage differences between factual and counterfactual conditions, not a continuous rate per degree. Please consider revising the labeling or adding an explanation to properly contextualize these graphs.
7. The inclusion of p-values in your graphs is understandable for visual emphasis; however, since you are comparing two distinct physical systems with comparable dynamics, the p-value may not be needed (Shepherd, 2021). It might be worth reconsidering their necessity.
8. Please provide explanations for each of the terms used in Equation 2 within the main text
9. Since the Methods section functions as a kind of prologue in the current structure, it would be helpful to reintroduce key abbreviations here.

Results

The results section is well-structured and generally easy to follow. I appreciate the consistent layout across subsections, which supports clarity and helps guide the reader through the analysis. The results are elaborate and clearly presented, and the interpretations are well connected to the data. That said, I find that in some places the discussion would benefit from more nuance—particularly by acknowledging the limitations of either your experimental setup or situating the findings more explicitly within the broader scientific context. Below, I offer some general remarks, followed by more detailed comments on each subsection.

One structural element I missed in the figures is a map showing the difference between the Factual and Counterfactual worlds. Such a map would immediately convey where and how the largest differences occur. I can see that you’ve intentionally kept the layout consistent across subchapters, which supports readability. However, some figure panels currently lack labels, which can be confusing. I understand that adding a difference map and labeling every panel presents some technical challenges, but I believe these additions are necessary. If you’re open to restructuring, one possible approach would be to split

the visuals into two sets: (1) maps, and (2) statistical summaries (e.g., line or violin plots). Since these convey different aspects of the results, separating them may even improve clarity. This structure would also allow you to repeat the legend in the distribution plots—currently missing—which would spare the reader from having to search through earlier panels to interpret the results.

The figure captions could benefit from being more explanatory to help guide the reader through the interpretation, especially for those skimming the figures without reading the full text. For example, the term "Theoretical Quantiles" is used without clarification—this concept is not widely familiar and would benefit from a brief explanation of what it represents and why it's used. If space is a concern, one option would be to provide a more detailed description in the caption of Figure 2, and then refer back to it in the captions of subsequent figures. This would balance clarity with conciseness across the manuscript and will aid in understanding important graphs that are perhaps not intuitive.

A second point relates to your use of the Clausius–Clapeyron (CC) rate (7% per K) as a benchmark throughout the results figures. While this reference makes sense for variables like precipitation and precipitable water, applying it to other variables—such as CAPE, diabatic heating, or maximum graupel—results in a mismatch. These are not expected to scale with CC in the same way, making such comparisons misleading. As shown in Figure 5 of Romps (2016), CAPE begins to diverge from CC scaling above ~ 300 K. This highlights the importance of including temperature information earlier in the manuscript, as noted in my comment on the introduction. Without that context, it's difficult to assess whether CC scaling is even applicable in your case. You also reference Pojol et al. (2021) to support the comparison, but I couldn't find this citation in your reference list, nor locate it independently. Unless there is a strong justification, I would recommend removing the CC reference line for these variables. Instead, comparisons to known or expected climate change responses specific to each variable would be more meaningful and would strengthen the interpretation of your results.

Changes in rainfall intensity and area

L123 The grammar here is a bit awkward. Please remove 'notably' and insert 'well' after 'distribution of precipitation' to improve readability.

L124 See the earlier comment on the introduction regarding Figure 1b—its interpretation is currently unclear and may benefit from some redesign for clarity.

L127 To ensure consistency in terminology, replace 'present-day' with 'factual'. You could also improve the flow by introducing this analysis with a sentence such as: "When comparing the factual and counterfactual simulations..."

L134–137 The language used here is quite strong considering the considerable uncertainty in the results. It would be helpful to present the full context rather than focusing only on the appealing outcomes. In the figure, the P99 does not appear to add much value—do you need it to support your main argument? If its inclusion is intended to highlight the limitations of your analysis (i.e., that the noise or uncertainty may exceed the signal), that would be a valid point, but please clarify this explicitly in the text. That said, I do appreciate the use of the AEMET data—it provides a valuable link to operational relevance.

L140–143 You refer to 'similar studies' (plural) but cite only one. While it's not necessary to elaborate on each study's findings, consider adding two or three relevant references after "... findings from similar studies." You can then remove the following sentence for a more concise presentation.

L144–150 This is a strong and thoughtful analysis of quite interesting results. It would be even more compelling if supported by a factual–counterfactual difference map (see earlier comment). I would also recommend avoiding the term 'violin plot' here—besides the need for a proper panel label, the figure shows a distribution. It's better, in my opinion, to refer to the content (e.g., distribution of values) rather than the specific plotting method.

Changes in moisture content and fluxes

L169 Consider starting the paragraph with a reference to Figure 3a. This would guide the reader toward the relevant visual right away, rather than waiting until L172.

L173 You choose for 'counterfactual' earlier in the paper, please use consistently and replace 'pre-industrial'.

L185–186 You might consider replacing 'In contrast' with 'As expected', though this depends on your interpretation. Given the thermodynamic differences in your experimental setup, such results could be anticipated. I'll leave the choice to your judgment. Also, I suggest revising 'notable PW reduction' to 'notably lower PW' for clarity and smoother phrasing.

L211–220 This section is an exact duplicate of the section before it, please delete.

Changes in physical mechanisms controlling extreme rainfall

L222 The word 'rainfall' is uncountable and does not take a plural form. Please remove the 's' at the end of the section title.

L226–227 The sentence is unclear and difficult to follow. Please rewrite.

L231–234 You might consider referring back to the CAPE results in combination with the updraft analysis. This could help provide a more complete and convincing interpretation of the findings presented here.

L243–255 The argument presented in the graupel paragraph would benefit from additional nuance. While it is well established that deep convection can enhance graupel formation through stronger updrafts and increased riming (e.g., Rutledge and Hobbs (1983); Tao et al. (2012)), the link between higher graupel content and increased rainfall rates is not straightforward. This relationship is influenced by microphysical parameterizations and storm dynamics, and remains highly model-dependent (e.g., Milbrandt and Yau (2005); Morrison et al. (2009)). In some contexts, graupel may enhance precipitation efficiency through melting or collision-coalescence, while in others it may indicate inefficient fallout or hail production. As such, additional justification and context are needed before drawing firm conclusions about the role of graupel in amplifying precipitation rates. The claim that increased graupel combined with stronger updrafts leads to super-Clausius-Clapeyron scaling requires more justification, as current literature shows this outcome is conditional and context-dependent (e.g., Singleton and Toumi (2013); Muller and Takayabu (2020)). Microphysical enhancements like graupel can influence precipitation intensity, but super-CC scaling also depends on storm structure, CAPE, and precipitation efficiency.

Discussion

In the discussion section, you bring together the individual variables and interpret them in relation to one another, which is valuable and strengthens your narrative. However, this integrative analysis could be expanded further to deepen the reader's understanding of how the variables interact within the system.

More importantly, the discussion currently lacks a critical reflection on the limitations of your modeling setup and the nuance or uncertainty associated with your findings. Including a paragraph that openly addresses these limitations—such as model resolution, parameter choices, handling of dynamics, or the representativeness of your results—would improve the transparency and credibility of your conclusions.

Another important element that could strengthen your discussion is situating your results more explicitly within the context of existing literature. Are your findings consistent with those from previous attribution or PGW studies in the Mediterranean region or other flash flood events? For example, how do your estimates of precipitation amplification compare to similar analyses (e.g., Zappa and Shepherd (2017); Orth et al. (2021))? Explicitly highlighting whether your results confirm, extend, or diverge from these earlier studies would help position your contribution more clearly within the field.

This also provides readers with a sense of how robust or novel the physical mechanisms you identify may be across different events and modeling approaches.

L273–275 Please clarify that your study does not provide evidence for changes on a global scale. I understand that this is not your intention, but the current phrasing could be misinterpreted. Consider rewording to clearly position your findings within the context of a regional study, while linking them to broader global trends in flash flood occurrence.

L282–286 This is indeed the appropriate place to synthesize the various variables into a collective interpretation of your results — something you could do even more extensively. That said, this paragraph currently lacks nuance, context, and critical reflection on the limitations of your setup. While it is valid to explore the potential for super-Clausius–Clapeyron (CC) scaling using the variables you’ve analyzed, the paragraph frames this as the primary goal, whereas your results offer more than just evidence for super-CC behavior.

Additionally, you argue that increased WVFlux underscores enhanced moisture transport rather than wind changes, but it is unclear how this conclusion is derived, given the limited description of how dynamics are treated in your model setup. Presumably, the use of upper-atmospheric nudging means that large-scale wind patterns are constrained, allowing you to attribute WVFlux increases primarily to humidity. However, nudging divergence may still influence vertical motion via the continuity equation. It would strengthen your interpretation to discuss how this nudging approach affects updrafts and what this implies for your conclusions.

Finally, your mention of “non-linear processes” returns to the earlier concern about the appropriateness of expressing your results in percent per degree Kelvin (%/K). A discussion of this metric’s limitations — especially when applied to quantities like CAPE or graupel — would be valuable here.

Figure 5 This is an excellent figure that effectively synthesizes your key findings. It deserves more emphasis in the manuscript. I would suggest positioning it as a highlight or centerpiece figure.

Final remark

Taking the feedback into account, I believe this paper will make a strong contribution to the field. It has been an honour to be involved in the process. I am not a fan of single-blind reviewing and will therefore sign my name.

Linda van Garderen

References

- Dare, R. et al. (2019). Attribution of flash flooding events using a storyline approach. *Journal of Hydrometeorology*, 20(6):1234–1246.
- Milbrandt, J. A. and Yau, M. K. (2005). A multimoment bulk microphysics parameterization. part ii: A proposed three-moment closure and scheme description. *Journal of the Atmospheric Sciences*, 62(9):3065–3081.
- Morrison, H., Milbrandt, J. A., Thompson, G., and Bryan, G. (2009). Impact of cloud microphysics on the development of trailing stratiform precipitation in a simulated squall line: Comparison of one- and two-moment schemes. *Monthly Weather Review*, 137(3):991–1007.
- Muller, C. J. and Takayabu, Y. N. (2020). Response of precipitation extremes to warming: What have we learned from theory and idealized cloud-resolving simulations, and what remains to be learned? *Environmental Research Letters*, 15(3):035001.
- Orth, R. et al. (2021). Using storylines to understand the impact of climate change on flash floods in the european alps. *Journal of Hydrometeorology*, 22(3):1233–1245.

- Romps, D. M. (2016). Clausius–clapeyron scaling of cape from analytical solutions to radiative–convective equilibrium. *Journal of the Atmospheric Sciences*, 73(9):3719–3737.
- Rutledge, S. A. and Hobbs, P. V. (1983). The mesoscale and microscale structure and organization of clouds and precipitation in midlatitude cyclones. viii: A model for the “seeder–feeder” process in warm-frontal rainbands. *Journal of the Atmospheric Sciences*, 40(5):1185–1206.
- Schär, C., Frei, C., Lüthi, D., and Davies, H. C. (1996). Surrogate climate-change scenarios for regional climate models. *Geophysical Research Letters*, 23(6):669–672.
- Shepherd, T. G. (2014). Atmospheric circulation as a source of uncertainty in climate change projections. *Nature Geoscience*, 7(10):703–708.
- Shepherd, T. G. (2016). Storyline approach to the construction of regional climate change information. *Proceedings of the Royal Society A*, 472(2186):20160145.
- Shepherd, T. G. (2021). Bringing physical reasoning into statistical practice in climate-change science. *Climatic Change*, 169(2):2.
- Singleton, A. and Toumi, R. (2013). Super-clausius–clapeyron scaling of rainfall in a model squall line. *Quarterly Journal of the Royal Meteorological Society*, 139(671):334–339.
- Tao, W.-K., Simpson, J., and McCumber, M. (2012). An ice-water saturation adjustment. *Meteorology and Atmospheric Physics*, 50(1-3):107–122.
- Zappa, G. and Shepherd, T. G. (2017). Storylines of atmospheric circulation change for european regional climate impact assessment. *Journal of Climate*, 30(16):6561–6577.

NCOMMS-25-38404

“Human-induced climate change amplification on storm dynamics in Valencia’s 2024 catastrophic flash flood” by Calvo-Sancho et al.

The authors have addressed most of the reviewers' comments, clarifying doubts and amending the text and/or figures to enhance the paper's quality and readability. I appreciate the improved connection to existing literature and for having better emphasized the novel aspects of their work, particularly those related to the sub-daily scale, which I find the most interesting. I have a few more comments for the authors which do not affect my recommendation to the Editor that the article is eligible for publication, as it can contribute significantly to understand the role of global warming in the intensification of extreme precipitations.

Regarding my comment on Figure 1, as I explicitly wrote in the previous review, I did not ask to add any figures or further formal analysis to be shown, but just a summary comment to assess the ability of WRF to reproduce correctly or as fair as possible the real event evolution, which would indirectly add robustness to the conclusions.

Extended Data Figure 7: I do see that “precipitation efficiency is the ratio Precipitation/Condensation”, the comment was relative to the x-axis tick labels of the PDFs; the clarifications for previous and this figure captions have anyway clarified my concern.

L116 “in a matter of hours” might sound better “within few hours”

Figure 1 caption: (a) Geopotential height ;

In general I would use the verb “to highlight” instead of “to underscore”

L 238-239: The lack of change in the wind field is in part due on the experiment set-up (as explained in the Limitations section), which does not allow to draw conclusions about their role in the changes of the event respect with pre-industrial climate. On the other hand the set-up itself allow to isolate the role of heat and vapor in modulating the event intensity.

Extended Data Figure 8: I prefer the version of this figure in the reviewers’ report, with also differences between factual and counterfactual profiles are plotted.

Second Review: Human-induced climate-change amplification of storm dynamics during the 2024 catastrophic flash flood in Valencia

I thank the authors for their detailed response to my first review and for the revisions that further clarified the manuscript. In my view, the paper is close to being ready for publication. I only have a few minor comments.

1) Presentation of difference maps

Thank you for your explanation regarding the figure structure. I fully understand your rationale for organizing the visuals by variable, and I agree that presenting spatial patterns alongside their statistics can support an integrated narrative — this is a reasonable and defensible choice.

That said, even within such a structure, the absence of a difference map still places interpretative effort on the reader. While the differences can indeed be spotted by comparing the factual and counterfactual maps, asking the reader to do this work reduces clarity and weakens the immediacy of your message. A difference map would make the key signal explicit and strengthen the results, rather than requiring the reader to extract it manually.

I do not insist on a specific solution, but I would encourage reconsidering. If space is a concern, one option could be replacing the factual/counterfactual pair with a single difference map. Alternatively, adding the difference map into the existing panel (or as supplementary material) would also solve the issue without changing your narrative structure.

To be clear, this remains a recommendation — the final choice is yours. My view is simply that providing a difference map in some form would make the manuscript stronger and more reader-friendly.

2) Lines 122–124: interpretation of Zappa & Shepherd (2017)

The manuscript currently cites Zappa & Shepherd (2017) as supporting increased extremes in Mediterranean winter precipitation. Their findings instead indicate a reduction in winter precipitation over the Mediterranean under climate change and do not provide evidence for increased extremes. Please revise the text to reflect this and situate your results within that context.

3) Lines 141–144: vertical mismatch of the storm

Thank you for clarifying in your response that you chose to let the atmosphere freely respond, given the strong lateral boundary conditions. This is a reasonable modeling choice, and the results indeed remain convincing and well-interpretable.

At the same time, this decision likely contributes to the slight vertical shift observed between MSLP and GPH-Z500. Since vertical gradients are central to your analysis, it would be useful to briefly acknowledge how this mismatch arises from the chosen setup and to discuss its potential influence on the results. Although the overall atmospheric dynamics remain comparable, the fields are not perfectly aligned — which may influence the location and intensity of precipitation — and explicitly noting this would help contextualize the interpretation for the reader.

A short paragraph in the Supplementary Material outlining this point (and why, despite it, the comparison remains valid) would be valuable for readers interested in methodological rigor

and model behavior. It would also pre-empt questions about the implications of allowing free atmospheric adjustment. In my view, this addition would not complicate the narrative but would strengthen confidence in the approach and transparently frame one of its limitations.

Thank you for your hard work — I look forward to seeing this paper published.

Linda van Garderen

Reviewer #1 (Remarks to the Author):

Review for: Climate change unleashed: physical-based attribution analysis proves human-induced amplification of Valencia's deadly flooding Calvo-Sancho et al.

This study attributes the influence of climate change on the extreme rainfall event that occurred near Valencia on 29 October 2024. The event was truly remarkable and caused substantial societal impact. The study is timely and, whilst other groups have made similar attempts to attribute the influence of climate change on this event, the novelty of this work is in probing the convective scale dynamics of the storm and the role of microphysical processes.

The manuscript is concise and well written (with the exception of some missing detail described below), and the figures are clear and generally appropriate. My only methodological concern relates to the robustness of the results as I describe below. If the authors are able to address this concern, along with my more minor comments on the presentation, I would be very happy to see the work published.

We are very grateful to the reviewer for their positive evaluation of our work and for the considerable effort devoted to improving its quality and scientific impact.

Major comment

- The authors compare one simulation of the Valencia rainfall event (the “factual” simulation) with an ensemble of pseudo-global warming simulations (the “counterfactuals”) produced by modifying the initial and boundary conditions with the aim of removing the influence of the warming experienced since preindustrial times. The ensemble of counterfactuals is generated by taking changes in temperature and humidity between pre-industrial and present day periods from a set of CMIP6 simulations. As such, the method aims to sample uncertainty associated with the amount of climate change experienced between pre-industrial times and the present day. This method will also, perhaps unintentionally, act to sample some variability associated with the chaotic evolution of the atmosphere in the same way that an ensemble NWP forecast does. This is useful as it adds to the robustness of the results by sampling a range of possible outcomes. My concern, however, is that only one factual simulation is considered, and furthermore, that this factual simulation has apparently been hand-picked from several simulations because it best reproduces the extreme high rainfall anomalies observed (see Methods: “Several different configuration tests have been carried out to obtain the most realistic factual simulation.”). Given this, I am left unsure how robust the results are to the choice of factual simulation used. Can the authors provide reassurance that the results are robust in an ensemble forecasting sense? One way to do this could be to run a set of simulations starting at different initialisation times in the run up to the event. Do they all exhibit the same changes relative to the counterfactuals?

Thank you for this insightful comment. We fully agree that the robustness of the results would be strengthened by considering an ensemble of factual simulations rather than a single realization. However, the PGWA usually uses the simulation which reproduces

better the real event in the perspective to attribute it to climate change. This is because if the NWP model does not fairly reproduce present climate events, it would not be able to conduct attribution experiments. To address this concern, we have performed additional experiments with different initialization times (October 28th 00, 06, 12UTC, and 29th 00UTC) for both the factual and counterfactual configurations. All initializations exhibit similar relative changes between factual and counterfactual runs, which provides confidence that the main conclusions are not sensitive to the specific initialization chosen (see Fig. RR1).

Figure RR1: 24-h total accumulated precipitation in factual (left column) and counterfactual (right column) in different initializations.

In the manuscript, we used the factual simulation (1800UTC on 28 October 2024, see Fig. RR1) originally selected because it reproduces the observed event most realistically. However, to increase transparency and robustness, we have now included in the Methods section a statement clarifying this procedure, and we provide in the Supplementary Material the 24-h accumulated precipitation maps for each initialization, allowing direct comparison across the different runs.

Minor comments

- The figure captions lack important details. Please review these to ensure they fully describe the figures, paying particular attention to:

o Figure 1 caption: What is the accumulation period for the rainfall map?

We have added the accumulation period (24h). Thanks.

o Figure 2 caption: Panel a: Are these distributions a single time or several (if so, what time(s)?). What does ‘theoretical quantile’ mean? Panel b: Please describe what is shown here (I think I understand, but only after reading right to the end of the Methods section – it is not clear from the caption). Panel c: The meaning of this sentence is not clear, how about “Percentage increase in the total area exceeding thresholds equal to the P90, P95 and P99 percentiles from the factual simulation and the AEMET red warning in the Valencia region.” Panel d: Please emphasise for clarity that the factual panel is a single simulation but the counterfactual is the mean of the ensemble of simulations. What is the p-value showing here?

Thank you for these points to improve the readability of the manuscript. The distributions are for the whole storm period (we clarified this in the methods and in the caption). We also amended Panel B, Panel C and Panel D captions. The theoretical quantile means the expected quantiles values under the fitted reference distribution which are compared with the empirical quantiles of the observed data (i.e., the 1-hr rainfall rate data in the Valencia region).

o Figure 3 caption: Do you mean convective available potential energy? What time is shown in all of the panels? Are the PDFs constructed from a single time? What does “non-climate CMIP6 model forcing” mean? You say the legend “shows statistical significance”, but of what? Please spell this out, e.g. “shows the p-value from a Mann-Whitney U test which aims to assess whether there is a significant difference between the two distributions”.

Thanks for your suggestions. Indeed, the word ‘available’ was missing in the caption title. Additionally, the panels show the average (added in the caption), and the PDFs are for

the whole period. Additionally, the term "non-climate CMIP6 model forcing" has been modified to "anthropogenic forcing."

We have rewritten the figure caption to clarify the Mann Whitney U test.

o Figure 4 caption: "Human-induces" -> "Human-induced". Panel a: What is maximum diabatic heating? Is this the maximum in the vertical, or in time. If in time, what level is shown? If in the vertical, what time is shown? Panel b: "Maximum updraft" -> "Maximum updraft speed". What does "non-climate CMIP6 model forcing" mean? All three PDFs look to only show values above a certain threshold, please specify what this is. I find it very hard to interpret what the p-values mean in this case because you are asking if the shape of the tail above the threshold is different between the different simulations. Would you expect it to be? Surely a more obvious question is how many more data points lie above the threshold?

Thanks for your observation. We have amended this error. Panel A shows the time-mean of the maximum diabatic heating in the vertical column.

The threshold is the same in both the factual and all counterfactual simulations, and it is defined as the minimum value along the x-axis for each parameter analyzed in this Figure (0.1 K s^{-1} , 20 m s^{-1} , 10 g kg^{-1} , respectively). We introduce these thresholds in order to focus the analysis on larger values, since these are the ones that drive extreme precipitation events, such as the Valencia event. As shown in Figure 4, both simulations include several near zero values, but the differences that are most relevant for understanding how climate change enhanced the Valencia event are in the upper tail of the distribution. Regarding the p-values, our goal is to test whether the shape of the upper tail (above the defined threshold) differs between factual and counterfactual climates, as this is where the potential signal of changes in the drivers of extreme precipitation is most likely to appear.

The number of data points exceeding each threshold in the factual simulations is 646 for diabatic heating, 4086 for maximum updraft speed, and 1351 for maximum graupel in column. In the counterfactual simulations, the corresponding mean values are 365, 4023, and 1394, respectively.

- The description of the simulations lacks some important details:

o What domains were used for the 3km and 1km nesting?

We use this WRF nesting configuration. We included the domain figure in Supplementary Material (Fig. S2). Thank you for this point.

o What time were the runs initialised and how long were they?

Thanks. We added this information to the methods of the revised manuscript.

o Has the performance of this version of the model at simulating extreme convective events been evaluated? Please provide relevant literature.

To the authors' best knowledge, this specific WRF version (v4.5) has not yet been evaluated at simulating extreme convective events. However, we consider that the evaluation of model performance is more closely related to the choice of the physical parameterizations than to the model version itself. In this regard, the physics configurations adopted in this study have been applied and assessed in the analysis of several extreme convective events, for example, within our group research, in tropical transitions events where deep-moist convection plays a central role. This is a sample of published studies that we used this physics configuration: <https://doi.org/10.1002/qj.4523>, <https://doi.org/10.1016/j.atmosres.2023.106801>.

- An important caveat to this work is that all the results presented are conditioned on the same large scale dynamical setup occurring in different climates. If the frequency of occurrence of such setups changes (or their dynamical intensity changes) then these will have important impacts which are not considered here. Whilst this is well known within the attribution community, I would strongly encourage the authors to be explicit about this point for clarity. Can they draw from the literature or IPCC assessment reports to comment on the state of understanding and uncertainty in this respect for these types of events? At the very least, the caveat should be mentioned in the abstract and the conclusions.

Thank you for this helpful recommendation. We agree that the paper needs a limitation statement with a critical reflection on the limitations of our approach. In the Methods section we have added a subsection assessing the limitation regarding the modeling setup, PGWA and change per degree warming concern as you point us above. In addition, we

added a brief sentence in the introduction about IPCC assess these events and the uncertainty to establish changes [L66-69].

Typos / points of clarification

Abstract: Several acronyms are used in the abstract without being introduced. Please define these.

Done, thanks.

L43: “in Turis official weather station” is not good English. How about “in the official weather station at Turis”? Also, over what period was the 771.8mm accumulation?

Thanks. It was amended.

L43; “and record-breaking in one hour in Spain” does not make sense. How about “and breaking the record for one hour rainfall accumulation in Spain”?

It was amended. Thanks.

L45: It is worth saying the WRF model is run at km-scale resolution.

Done, thanks.

L46: “sourced with CMIP6 models” is unusual terminology. Please either expand to say in what way CMIP6 models are used here, or (probably better) just remove the phrase and leave the details to the main text.

This phrase was rewritten. Thanks for your suggestion.

L48: “Observed climate change led to...” is not accurate. You explicitly use simulated changes in climate to drive the model. How about the more precise 'Compared to pre-industrial conditions, present-day conditions lead to...'.

Thank you for your suggestion. It was amended.

L49: In what way is 20%/C consistent with Clausius-Clapeyron? Please expand.

Thank you for this point. It was a mistake and we want to say that present-day climate conditions exceed the Clausius-Clapeyron scaling (+7%/°C). We amended it in the revised manuscript.

L51: You say “stronger vertical velocities” but do you also mean larger CAPE and stronger diabatic heating?

This phrase was rewritten. Thanks.

L64: Presumably breaking Rossby waves are only relevant for extreme precipitation events in the extra-tropics?

We thank the reviewer for this insightful comment. We agree that Rossby wave breaking has been mainly identified as a key dynamical driver of extreme precipitation in mid-latitude and extratropical regions (Woollings et al., 2023 -cited in the manuscript-).

However, recent studies have shown that this process is not confined to the extratropics, as it can also favor extreme rainfall in arid and semi-arid regions worldwide (De Vries et al., 2024 -see citation in the main text-). In the Mediterranean, Rossby wave breaking and the associated development of cut-off lows are well-documented mechanisms triggering heavy precipitation events (Mishra et al., 2025, -see citation in the main text-). Therefore, while in our case study thermodynamic drivers (enhanced moisture, convective instability, and cloud microphysics) are more determinant, we referred to Rossby wave breaking as it remains a relevant dynamical process highlighted in the literature to explain the recurrence of extreme precipitation both in the Mediterranean and in other extratropical regions.

L66: “de Vrier” -> “de Vries”

Done. Thanks.

L96: Missing word near “atmospheric river-like from”?

Amended. Thank you.

L110: “using the physical-based attribution approach” Please add a sentence or to outline the main differences between the factual and counter-factual simulations, and refer the reader to the methods section for full details.

Thank you for this point to improve the readability of the manuscript. We have added a sentence with the main differences of factual (present-day climate simulation) and the counterfactual (pre-industrial-like climate simulation, i.e., the Valencia event forcing removing the human-induced climate signal from the initial/boundary conditions) world.

L132: Are the areas smaller for any one member? You use a log scale so negative values wouldn't show up if they were.

As shown in Extended Data Table 2, only one member (INM-CM-4-8) shows a simulation of decreasing areas (-13%) in the >300 mm threshold. Thanks.

L211: This whole paragraph is a repeat of the previous one.

Fixed. Apologizes and thank you for this observation.

L226: “of diabatic heating more intensity” does not make sense.

This sentence was rewritten. Thanks.

L227: “The median diabatic heating increase from pre-industrial to present-day climate is 29.5%”. What does this mean? Is this out of all grid points with non-zero diabatic heating?

We have revised the wording to enhance clarity and ensure this point is more easily understood in the text. The increase is +29.5% in the median value compared factual

against counterfactual. It can be observed in the first PDF of Figure 4. Thanks for your comment.

L317: What does “precipitation numerical resolution” mean?

Fixed. We removed 'numerical' because it was included by mistake in the draft. Thank you.

L338: “we had to concatenate” -> “we concatenated”.

Done. Thank you for this observation.

L347: As understand it, there is only one factual simulation, but you say “are compared against factual simulations”. Please clarify.

Thank you for this point. It was amended in the revised manuscript.

L386: What is “the storm period”?

Thank you for this point. The storm period in the simulation is 13 hours, 29 October from 04 to 16 UTC.

Extended Data Figure captions 3-7: Shouldn't these say “As in Extended Data Figure 2”?

Thank you. It was amended.

Reviewer 2: (Remarks to the Author):

NCOMMS-25-38404

“Climate change unleashed: physical-based attribution analysis proves human-induced amplification of Valencia's deadly flooding” by Calvo-Sancho et al.

The authors attempt to frame the Valencia extreme event occurred in October 2024 in the context of impact of global warming. They use the model WRF to simulate the event, presenting a very quick assessment of the precipitation field against the rain gauges.

They then perform a comparison between the simulated event and a “synthetic” preindustrial scenario (built through CMIP6 models climate forcing with a Pseudo-Global Warming approach), quantifying the impact of the global warming on the intensity of the catastrophic event, through the study of the change of different storm drivers, and so claimed as “physical-based” attribution through the approach of the storylines.

Although the method is already known, the 2024 Valencia extreme event considered is still poorly studied, as well as its attribution to global warming. Some interesting aspect arises, in terms of climate change impact, considering the sub-daily scale and driver variables unusual in attribution studies; apart these, the novelties the author intend to share with the scientific community for advancing knowledge are hard to be captured and should be better highlighted. The comparison with existing literature should be enriched and the Method section should be improved (details on simulation configuration should be added or model namelists should be available to make the experiments reproducible). Some results should be discussed more in depth than simply reporting the plot evidences. Some of the cited papers are missing in the bibliography and a revision of English is needed.

Below some specific comments which I think can help the authors to improve the paper, which can be considered for publication after major revisions.

Thank you for your time and effort in this insightful revision in our manuscript. Your comments have been very useful in improving the paper.

Comments

L76-79 I would say that also the method of Faranda et al., 2024 is physical-based as it is based on different hazard variables over a statistical base (also weighting the role of low- frequency modes of natural variability); they also reported about your same event. Your method is related to the single event analyzed (through well known PGW technique). I would better highlight the novel aspects you are presenting here, as well also discuss those ones that can eventually be generalized in the framework of climate change impact on wet hazard. Apart the specific event, for example I have appreciated that you have considered also the sub-daily scale, which is not as usual in attribution studies, but it has been suggested to be intensifying due to warming more sensibly than longer time-scale extremes (<https://doi.org/10.1029/2020GL089300>).

Thank you for this insightful comment to improve our manuscript, we really appreciate it. We agree that the ClimaMeter approach (Faranda et al. 2024) is conceptually based on physical

variables and, at the same time, considers the influence of low-frequency modes of natural variability, which justifies referring to it as physical-based method attribution. Conversely, Faranda et al. (2024) explicitly acknowledge a methodological limitation of ClimaMeter: “*we do not estimate quantitatively the effects of climate change and natural variability on the physical characteristics of a given extreme event, contrary to probabilistic approaches for extreme event attribution*”; in addition, this approach depends on circulation analogues derived from reanalyses and is less suitable when local processes dominate. Therefore, we see our PGW-based experiment as complementary to ClimaMeter: by forcing the boundary conditions of the numerical model, we obtain physically consistent counterfactual conditions for the specific event, allowing a more direct quantification of climate change impacts on its physical features in the Valencia event.

We highlight in the introduction of the revised manuscript the novel aspects of this study, especially the focus on the sub-daily changes in rainfall intensity [L92-99].

L110 and L123 (and following) The authors introduce the “counterfactual” event, i.e. the event in a pre-industrial climate, then using it for comparison with the factual extreme in the following sections. I suggest a concise mention on how it is built in the “Introduction” or in the “Results” paragraph, then referring the following “Methods” section for a more in depth explanation.

Thank you for this point. We added more in-depth explanation in Methods and we refer to Methods section in Introduction [L129-136; 355-357].

Figure 1 It would be useful to add in panel “b” locations of sensors in panel “c” (Turís and Poio ravine); panel a: what is the data source? Moreover, the evaluation analysis against observed precipitation is very quick and based only on total precipitation map. The reader should trust that the event is overall fairly reproduced based on this plot (1b). The discussion here can be improved, even not necessarily adding further graphics but expanding the evaluation over other aspects about the storm reproduction against observations, which is not clear if has been performed (and not shown) or it is strictly limited on what shown on Figure 1.

Thank you for this point to clarify where both sensors are located. The data source from panel “a”) and the locations of sensors to panel “b”) were added. We agree that a deeper evaluation of the factual simulation against observations (official and personal weather stations) is needed. Considering that the main goal of the paper is to assess the role of ACC in enhancing the Valencia floods, we think that Figure 1b provides a sufficiently clear visual comparison against observed precipitation. This allows the reader to identify the westward displacement of the storm and its overall fair agreement with observational data from AEMET, AVAMET, and SISRITEL weather stations.

Section “Changes in rainfall intensity and area”: It would be good to have also a comparison (at least as discussion) between the two events large scale conditions.

Thank you for this point in improving the paper. Following the analysis of the Figure reported below, we added a short sentence about the comparison between the two large scale conditions at the start of the revised section “Changes in rainfall intensity and area”, L144-147. “*Large-scale conditions present some differences between factual and counterfactual simulations in the 500-hPa geopotential field (Extended Data Figure 9). In a preindustrial-like climate, the cut-off low would have been slightly deeper than in the present-day climate, but with a similar location. Regarding the MSLP, there are not significant differences neither in position nor intensity.*”

Fig RR1: Comparison between factual and counterfactual simulation of (a) 500-hPa geopotential height (m) and mean sea level pressure (hPa) at October 29th, 2024 12 UTC.

Figure 2, Some of the comparison criteria are explained in the figure captions but not in the main text L132-140 (for ex. “The total rainfall percentiles are calculated from the factual simulation”). Moreover some further discussion about how much your results here are in line with existing literature or not would help to better highlight any novel aspect arising (for ex., but not only, <https://doi.org/10.1016/j.wace.2025.100781>, <https://doi.org/10.1029/2024GL112392>, <https://doi.org/10.21203/rs.3.rs-2065981/v1>, doi.org/10.1007/s00382-023-06901-9, <https://doi.org/10.1029/2023GL105143>)

Thanks. Comparison criteria and more discussion was added to the main text to clarify [L161-163;L167-171], in particular in the Methods section. Moreover, additional references have been added in the revised manuscript.

L132, L286 confirming existing literature, that should be cited (ex. but not limited to: <https://doi.org/10.1038/nclimate2258>, <https://doi.org/10.1175/JCLI-D-16-0808.1>). I suggest to improve the discussion about these results.

Thanks. We added the suggested references.

L140 Any comment about the GCM giving larger extremes in the counterfactual event?

Thank you for your observation. It was amended since it erroneously referred to factual events.

L183 Figure 3: what is the vertical portion of the air column considered for PW and WVflux?

Thanks. It was added in the caption of Figure 3 for WVflux, for PW it is by definition the whole column.

L190 Extended data Figure 2: what is the level shown?

Thanks. The Extended data Figure 2 caption was amended.

L198-200 The link between the two statements and speculations, should be discussed more in depth.

Fixed. We discussed it in the same section [L227-234]. Thanks for this recommendation.

L201 and L204 zonal flux; is WVFlux integrated?

Yes, the WVFlux is integrated in the lower troposphere, between surface and 700 hPa.

L99, L206-207, L281, L299, L371, L388 citations missing in the references list.

Fixed. Thanks.

L226 (and following), L329, L330 citations missing in the references list or typo in referencing.

Fixed. Thanks.

L211-220 Copy of the previous paragraph.

Indeed. It was deleted. Thank you.

L238-39 You can speculate a bit here linking to a possible spatial increase of the risk/severity related to the increased storm area as well as downstream. Still in line with other literature about heavy storm sensitivity to global warming (ex. doi.org/10.1007/s00382-023-06901-9, <https://doi.org/10.1029/2023GL105143>).

Thank you for this point. We added more discussion, linking the increase of strong updrafts in a wider area (increased storm area) with a larger contribution to catchments.

Extended Data Figure 3 and 4: Check the captions. Moreover, is that one single level?

Checked and amended. It is the temporal mean of the parameter and the maximum value within the vertical column of this mean field. Thanks for this comment.

Extended Data Figure 7: are labels of violin plot x-axis right?

Yes, it is right. The precipitation efficiency is the ratio Precipitation/Condensation.

L282-84: From results I have understood that the increase of moisture is mainly contributing to the enhancement of its transport (due to almost unchanged winds), acting as primary driver of rainfall intensity. If I am correct, maybe this sentence needs some rephrasing.

Indeed. It was rephrased in the revised manuscript. Thanks.

Figure 5 caption: clarify “higher microphysics”. Changes in the figure are related to the Valencia case study and this should be specified.

Done, thanks.

L335 Please add details on how the GHG perturbation is built (or document from literature), so that your experiment is replicable.

We added more details about how the GHG perturbation is built. We modify the WRF source code and re-compile it to run the counterfactual simulations. Thanks for this helpful recommendation.

Minor comments

L60 The very first sentence of the introduction may sound misleading, although it is elaborated in the following lines. You may try to rephrase.

Rephrased, thanks.

L87 The term “overwhelming” is inadequate referred to natural hazards. Better using “devastating”

Thanks. It was amended.

L88 causing or triggering better than “promoting”

Thanks. It was amended.

L140 “Extended Data section”

Done, thank you.

L141 please cite properly. The DOI is not specified

Done, thanks.

L193-95 Review the English carefully

Thank you for your observation. It was amended.

L196 “a larger extent”: do you mean spatially?

Yes. It was amended, thanks.

L227 more intense

Modified, thank you.

L235-6 “dispersed” I guess you mean spatially; I suggest to link with following sentence (“, although” or similar) so that you can better highlight that this happens over fewer grid- points and ensemble members.

It was rephrased, thanks.

L247 “elevated” do you mean large values?

Indeed. It was amended, thank you.

L251 affecting

Done, thanks.

L261 produces

Done, thanks.

L327 I would not use PGW acronym before its extended form in the section title.

Done, thanks.

L348 is “surveys” here used as “research studies”?

It was amended [L407], thanks.

L385 only hourly precipitation?

It was amended [L446], thank you.

Reviewer 3: (Remarks to the Author):

Review: 'Climate change unleashed: physical-based attribution analysis proves human-induced amplification of Valencia's deadly flooding'

I would like to thank and congratulate the authors for their hard work and for contributing to the growing body of knowledge in extreme event attribution science. Out of appreciation for the effort invested in this study, I have reviewed the manuscript with care. The analysis shows clear potential, and with revisions to improve clarity and completeness, it could make a valuable addition to the field.

Goal and Summary of Article

The primary objective of this paper is to attribute the role of anthropogenic climate change in the meteorological conditions that led to the Valencia flash flood of October 2024. To achieve this, the authors employ a storyline-based attribution approach using a Pseudo-Global Warming (PGW) framework within a regional climate model, driven by ECMWF-IFS reanalysis data. The counterfactual (pre-industrial) simulations are constructed by modifying the boundary conditions with multi-model mean changes derived from CMIP6 GCM data.

The attribution analysis focuses on changes in precipitation intensity and spatial extent, as well as shifts in key atmospheric moisture variables, including CAPE, precipitable water, and water vapor flux. Additionally, the study examines physical mechanisms such as maximum updraft and graupel formation. Many of these variables are assessed in relation to the Clausius-Clapeyron scaling rate of approximately 7% increase in moisture per degree Celsius of warming.

We thank Dr. Linda van Garderen for this insightful revision to improve our work and appreciate your recognition of the scientific significance of this to the attribution climate change field. Below, we provided detailed responses to your concerns.

Title

I'd suggest rethinking the title, as the phrase 'Climate change unleashed' feels a bit sensational for a scientific paper. Since this work is aimed at peers, a more neutral tone would likely strengthen its credibility. You could consider starting with 'Valencia's deadly flash flood' to immediately highlight the event and its relevance to attribution, without losing impact.

Thank you for your suggestion. We agree the initial title feels a bit sensational. We changed to **Human-induced climate change amplification on storm dynamics in Valencia's 2024 catastrophic flash flood** to cover its relevance to attribution and highlight the event without losing impact.

Abstract

The abstract appropriately sets expectations about the paper's content. However, some rephrasing could enhance clarity and better emphasize the novelty of the results. Currently, the consistency with Clausius-Clapeyron scaling is highlighted as a key finding, but this alone may not fully engage readers or convey the paper's unique contributions. It is worth highlighting in the abstract that the study employs a storyline analysis using the PGW method, as this clarification helps address potential questions from readers early on.

Thank you. The PGW method has been incorporated into the abstract.

L44-45 The sentence referring to the event as "the major weather event" may be perceived as insensitive given the fatalities involved. Consider revising to something like: "... resulting in fatalities and ranking as the most destructive event for ..."

Thanks. It was amended.

L49-52 Reword this section to more clearly showcase the novel aspects of your findings, rather than framing them in a way that might imply a lack of new insight. Also, since the paper focuses specifically on updraft rather than vertical velocity (despite their overlap), it would be better to consistently use the term

“updraft” to align with the paper’s terminology.

Rephrased. Thanks.

L54 The term “risk” was not analyzed in the study; please adjust the wording accordingly to avoid any misinterpretation.

The sentence was modified. Thanks.

Introduction

While the introduction reads well and provides a good overview of the event under study, several revisions are necessary to strengthen the manuscript.

The literature review is notably lacking in coverage of storyline approaches and the attribution of extreme precipitation events. Additionally, several key references are missing, which also applies to the methodology section. For instance, the manuscript discusses storylines and atmospheric uncertainty but does not cite foundational works by T. Shepherd (Shepherd, 2014, 2016). Similarly, although the study employs the Pseudo-Global Warming (PGW) approach, the original developer C. Schär (Schär et al., 1996) is not referenced. Furthermore, previous storyline analyses focused on precipitation in the Mediterranean region (Zappa and Shepherd, 2017) and flash floods (Dare et al., 2019; Orth et al., 2021) are omitted. Expanding the literature review to include these key contributions would provide readers with better context and emphasize the significance of the current research.

Thank you for this point to improve the quality of the paper. The storyline and PGWA foundational works are cited now in the Methods section when we introduce and explain in-depth the PGW method. Regarding the storyline analysis focused on precipitation and flash floods, we have only added Zappa and Sheperd (2017) work because we cannot find in different scientific literary sources the Dare et al. (2019) and Orth et al. (2021) works. On the other hand, we expanded the literature review and we include two insightful works (Munz et al. 2024; Thompson et al. 2025) to provide a better context about the European and Mediterranean precipitation changes in a warming climate [L122-128].

The detailed description of the event is appreciated and essential for establishing a baseline understanding. However, the manuscript lacks any presentation of temperature profiles. Since precipitation characteristics, CAPE, and Clausius-Clapeyron scaling are all strongly dependent on temperature, it is crucial to include temperature information. This could be visualized in various ways, but the observed temperatures, counterfactual simulations, and the differences between factual and counterfactual should be clearly illustrated both spatially and temporally—and potentially vertically as well—to fully support the analysis.

We thank the reviewer for this point. We agree that it is crucial to include temperature information due to the strong interdependence between precipitation, CAPE and Clausius-Clapeyron scaling. We visualized the differences spatially, temporally and vertically. Moreover, we included the wind and updraft speed vertical distribution. The 2-meter temperature difference between the two discrete physical states (factual minus counterfactual) displays a positive anomaly $+1-1.2^{\circ}\text{C}$ in almost all the Valencia region domain, except in its northwestern areas, where some negative anomalies are present (Figure RR1a-c). On the other hand, the mean difference between these two states is $+1.08^{\circ}\text{C}$, a value that we used to compute the change per $^{\circ}\text{C}$ warming in the parameters that we analyzed in this work. Regarding the temperature (and dew point) temporal evolution, the difference between factual and counterfactual simulations remains similar in all timesteps. Overall, both spatial and temporal temperature differences are inherited from the CMIP6 forcings.

Regarding the wind speed vertical distribution, there is not any notable change between the factual and counterfactuals simulations at any level. On the other hand, the specific humidity displays notable changes in low-mid levels and negligible changes in upper levels. These changes in low-mid levels promote the notable increase in CAPE and updraft speed in the present-day climate in comparison to counterfactual simulations.

Hence, we will include the following new Figure (Extended Data Figure 8) in the revised manuscript to contextualize the temperature, specific humidity and wind changes.

Figure RR1: 2-m temperature spatial distribution during the storm period in (a) factual, (b) counterfactual runs, and (c) the difference between (a) and (b). (d) 2-m temporal evolution in factual and counterfactual simulations. Vertical profiles of (e) temperature, (f) dewpoint, (g) specific humidity, (h) wind speed, and (i) updraft speed for factual (black line) and counterfactual (orange line) and their difference. Orange dashed represents \pm standard deviation of the 15 counterfactual simulations and grey dashed represents \pm standard deviation of the 15 differences between factual and counterfactual.

below some smaller details to look at:

L68 The term "triggering mechanisms" is introduced but never revisited or explained. Please clarify what you mean by this or consider rephrasing to avoid ambiguity.

Thank you for this point. By "triggering mechanisms", we refer to the forcings that initiate convection (e.g., orographic lifting, frontal boundaries, low-level convergence, large-scale forcing), which promotes precipitation generation. To avoid ambiguity in the reader, we have replaced the term with convective initiation

[L77].

L69 Are the results best described as "contrasting" or "uncertain"? If the uncertainty exceeds the signal, this might create a contrasting appearance. Consider adding a sentence to clarify this distinction.

It was amended. Thanks.

L78 Please add "of" between "because" and "ACC" for grammatical correctness.

Thank you for your observation. Done.

L79 Remove the phrase "In contrast," as it is unnecessary here.

Done, thanks.

L83-84 Reword this passage to clearly articulate the specific research gap your study addresses, avoiding vague or general recommendations.

We rewrite this passage to include the research gap that we address in our study. "*The current attribution studies that quantify the contribution of ACC to the intensity, frequency and extent of extreme precipitation events focus on daily-scale. However, the sub-daily scales -where convective processes dominate- are still poorly characterized. Herein, we address this gap by analyzing sub-daily observations and high-resolution simulations to quantify the contribution of ACC in the Valencia extreme rainfall event.*"

L86 Replace the word "accumulations" with "accumulative values" for precision.

Done, thanks.

L94 The current sentence is unclear—how exactly do quasi-stationary convective systems get "triggered"? Please clarify what action or change you are attributing to the trigger.

Thank you for this point. It was a mistake when we drafted the manuscript and we want to mention that large-scale and mesoscale set-up favored the convective instability and the formation of quasi-stationary convective systems [L107].

L95-97 The grammar here is awkward, making the sentence difficult to understand. Please revise.

It was revised. Thanks.

L105-108 Demonstrating consistency with Clausius-Clapeyron scaling could also indicate that an atmospheric river is involved. Have you investigated this possibility? Can you confirm or rule it out based on your data?

Thank you for this well-focused and insightful comment. Yes, we conducted preliminary tests to assess the potential influence of moisture transport through an event similar to what is commonly referred to as an "atmospheric river" (AR). The main challenge in this regard lies in the considerable uncertainties surrounding how such events should be defined, both in terms of morphology and the minimum integrated vapor transport (IVT) required. In our view, and following Lorente-Plazas et al. (2020) – <https://doi.org/10.1029/2019JD031280>– this represents a major limitation for detecting and studying these events in the Mediterranean region. Essentially, it is not feasible to apply fixed IVT thresholds similar to those used for the California coast, for example. However, when using extreme IVT thresholds relative to the regional climatology (e.g., based on the monthly 85th percentile), it is possible to identify such moisture transport structures. Yet, morphological requirements, such as minimum length, may still exclude structures that significantly affect the eastern Iberian Peninsula. Recently, Davolio et al. (2023) - <https://doi.org/10.1016/j.wace.2022.100542>- proposed an adaptation of an algorithm for the detection of ARs in the Pacific Ocean to the Mediterranean, taking into account the peculiar morphology of the basin.

Within this context, we would like to convey that this issue is complex and somewhat beyond the scope of our study, which is why we decided not to address it in detail. For instance, a comparison using different AR detection methods and different catalogs would reveal much greater variability in the potential events affecting Valencia than in other regions of the world. What is clear, however, is that the synoptic configuration on October 29 strongly favored strong moisture transport from the subtropical Atlantic into the western Mediterranean. Moreover, given the easterly advection of moist air in the boundary layer, this transport likely included not only Mediterranean Sea moisture (78.15) but also additional moisture uptake from the Atlantic (8.5%) (Huang et al. 2025).

We have attached a figure (Fig. RR2) illustrating this point: using our own tracking algorithm (it is not published yet, but it is based on Brands et al. (2017) -<https://doi.org/10.1007/s00382-016-3095-6>- methods) and applying a 85th percentile IVT threshold (approximately $150 \text{ kg m}^{-1} \text{ s}^{-1}$ at the end of the detected track), an AR-like event can indeed be identified. Nevertheless, as shown in Figure RR3 of this response, IVT values on October 29 were comparable to those observed in other cases that did not produce extraordinary precipitation. This suggests that moisture convergence and small-scale processes, such as deep convection, were crucial in this event, rather than large-scale moisture transport alone.

Your comment is highly relevant and could certainly inspire future research. However, we believe that addressing this topic in detail here would detract from the clarity of our main message. Thank you again for raising this important point.

Figure RR2. Integrated Vapor Transport (IVT) field on 29 October 2024 at 20:00 UTC from ERA5 reanalysis. The ERA5 grid cells along the Valencia coast are highlighted in red (used as reference for detection), with the yellow circle marking the grid point with the highest IVT, where the potential atmospheric river impacts the coastline. The yellow line represents the full trajectory of moisture transport meeting the detection criteria (IVT exceeding the October 85th percentile based on the 1991–2020 climatology).

Figure RR3. For the ERA5 grid cells representing the Valencia coast (highlighted as purple circles in Fig.

RR2), the panel shows the hourly evolution of Integrated Vapor Transport (IVT) during October 2024. The black line indicates the mean IVT, while the dark blue and light blue shaded areas represent the 10th–90th and 5th–95th percentile ranges, respectively. The red line marks the 85th percentile of IVT for this month across the corresponding ERA5 grid points.

L109-112 Please explicitly state that your analysis uses a storyline approach with a regional model under PGW settings. This will help readers better understand the results without needing to refer back to the methods section.

It was rewarded. Thanks.

Figure 1 Suggestion: Consider replacing the color scheme in panel (a) with temperature to enhance clarity. Additionally, the unit used in panel (c) (mm/10min) may be unfamiliar—if it is not commonly used, please convert it to a more standard unit.

Thank you for this point to improve the readability of Figure 1. Anyway, we changed the color scheme enhancing the geopotential height to clarify the cut-off low position. Regarding the units in panel (c), we consider keeping them since they are the Spanish Meteorological Office standard units for rainfall.

Figure 1b The panel currently does not effectively show the difference between observed and modeled data. Please revise the visualization to more clearly and intuitively display the comparison.

Thank you for your suggestion. We have carefully explored several alternative ways to improve visualization and highlight the differences between the observed and modeled data. Anyway, after testing these options, we believe that the current version offers the most balanced and informative representation.

Methods

The Methods section would benefit from reorganization to improve clarity and reader comprehension. Beginning with an overview of the storyline approach would provide essential context before delving into specific details such as datasets. Presenting a general introduction to the methodology first, followed by the technical specifics, would help readers grasp the overall framework and purpose of each component. As noted earlier, some important references are missing and should be included.

Additionally, there are several questions and ambiguities that arise from the current description of the methods, which are outlined below:

1. Could you elaborate on the selection process for the CMIP6 GCMs used in your study? Additionally, to what extent might your choice of models influence the results?

Thank you for your question. We selected all available CMIP6 GCMs that provided the required variables for both the historical period and the SSP4-2.5 scenario from the ESGF data node (<https://esgf-node.ipsl.upmc.fr/search/cmip6-ipsl/>), as it was available at the time to download CMIP6 climate models. The final ensemble includes 15 GCMs, which we consider to be sufficiently a large sample to capture a representative spread of model responses. This ensemble size allows us to account for model-related uncertainties and provides a robust estimation of the climate signal. We acknowledge that the choice of models can influence specific results, but using a multi-model ensemble helps to reduce individual model biases and increases the reliability of overall conclusions.

2. Please specify the model domain more clearly, ideally including a map that illustrates the grid layout.

We will upload in a public repository the WRF namelists in order that the experiment could be reproduced. We included the domain figure in Supplementary Material (Fig. S2). Thank you for this point.

3. Clarify explicitly that the ECMWF-IFS data used are reanalysis products.

It was clarified in the revised manuscript. Line 346: *Several prognostic variables from the ECMWF-IFS analysis, used as initial and boundary conditions are modified.*

4. Explain how atmospheric dynamics are handled in the model. For example, is the upper atmosphere nudged or otherwise constrained? If so, how might this affect analyses related to updrafts?

We agree on the importance of spectral nudging for constraining the large-scale conditions of long-term climate simulations.

In this study, however, the lead time is only 36-h, and the domain is strongly constrained by lateral boundary conditions from IFS analysis at 0.1° resolution every 6-h. We consider that additional spectral nudging is not required because it may dampen mesoscale variability. It is particularly relevant in our work because we focus on mesoscale sub-daily processes (e.g., convective organization and updrafts), which could be weakened if the upper atmosphere were strongly nudged.

In contrast, in climate-scale simulations spectral nudging is necessary to constrain the upper atmosphere and prevent large-scale drift. However, this is not the approach we adopted for the Valencia case, we simulated uniquely the 36-h period that covers the whole event.

Thanks for these helpful questions.

5. Indicate whether your analysis represents averages over the entire model domain and whether this approach is consistent across all variables.

In the method we clarified the domain used to analyze the different parameters in the Valencia event. On the other hand, the PDFs, violin plots, Q-Q plot and change per degree warming plots consider the data only in the Valencia region domain. Thus, we did not average over the entire model domain. This point has been clarified throughout the manuscript. Thanks for indicating this point.

6. Many variables are expressed as percentage change per degree of warming, which provides valuable insight. However, given that your study compares only two states differing by approximately 1.08°C , can you confidently generalize this as “per degree warming”? Without scenarios spanning a wider range of warming, the relationship beyond this specific temperature difference remains uncertain. While comparing to Clausius-Clapeyron scaling is valid for some variables, labeling changes as “per degree warming” might be misleading, as it implies a linear relationship irrespective of the warming magnitude. Strictly speaking, you are presenting percentage differences between factual and counterfactual conditions, not a continuous rate per degree. Please consider revising the labeling or adding an explanation to properly contextualize these graphs.

Thank you for this fair point. Although this study is based on two discrete physical states and does not explore a wider range of warming conditions, we do not agree that labeling change as “per degree warming” is misleading our message in the manuscript. Labeling changes as “per degree warming” simply expresses the relative change between the present-day and pre-industrial-like climate, which does not imply a linear relationship beyond this range. However, in the Methods section, we added a Limitations statement of this work where we included this problematic to contextualize our results.

Moreover, we base our argument on the CC relationship that extreme precipitation is likely to intensify with saturation vapor pressure at about 7 % per degree warming which has been found to be robust in many studies (e.g., IPCC AR6 2021; Fowler et al. 2021; <https://centaur.reading.ac.uk/95124/1/Fowler20NatRevV1A.pdf>) and is also robust for event based analysis (e.g., <https://royalsocietypublishing.org/doi/10.1098/rsta.2019.0546>)

7. The inclusion of p-values in your graphs is understandable for visual emphasis; however, since you are comparing two distinct physical systems with comparable dynamics, the p-value may not be needed (Shepherd, 2021). It might be worth reconsidering their necessity.

Thanks for your recommendation. However, we think that the p-value in the graphs provides statistical robustness to the results, and we decided to leave this information unmodified.

8. Please provide explanations for each of the terms used in Equation 2 within the main text

Thank you for this point. We added the meaning of each term in Eq. 2.

9. Since the Methods section functions as a kind of prologue in the current structure, it would be helpful to reintroduce key abbreviations here.

Thank you for your suggestion. We acknowledge that, in Nature Communications, the Methods section appears at the end of the article, unlike in other scientific journals where it is presented earlier. While this structure may lead to some acronyms being less immediately recognizable, we have reviewed several articles published in Nature Communications and found that it is common practice to define acronyms only once, typically at their first appearance in the main text.

Results

The results section is well-structured and generally easy to follow. I appreciate the consistent layout across subsections, which supports clarity and helps guide the reader through the analysis. The results are elaborate and clearly presented, and the interpretations are well connected to the data. That said, I find that in some places the discussion would benefit from more nuance—particularly by acknowledging the limitations of either your experimental setup or situating the findings more explicitly within the broader scientific context. Below, I offer some general remarks, followed by more detailed comments on each subsection.

One structural element I missed in the figures is a map showing the difference between the Factual and Counterfactual worlds. Such a map would immediately convey where and how the largest differences occur. I can see that you've intentionally kept the layout consistent across subchapters, which supports readability. However, some figure panels currently lack labels, which can be confusing. I understand that adding a difference map and labeling every panel presents some technical challenges, but I believe these additions are necessary. If you're open to restructuring, one possible approach would be to split the visuals into two sets: (1) maps, and (2) statistical summaries (e.g., line or violin plots). Since these convey different aspects of the results, separating them may even improve clarity. This structure would also allow you to repeat the legend in the distribution plots—currently missing—which would spare the reader from having to search through earlier panels to interpret the results.

Thank you very much for your suggestions. We understand the value of including difference maps and appreciate the idea of organizing the visuals into separate sets (maps vs. statistical plots). However, in our view, structuring the figures by variable—as we currently do—makes it easier for the reader to follow the story. We believe that showing both the spatial patterns and their corresponding statistics together helps to interpret each variable in a more integrated way, without having to jump between different figures.

Moreover, based on your comments, we have improved the labeling of all figure panels.

The figure captions could benefit from being more explanatory to help guide the reader through the interpretation, especially for those skimming the figures without reading the full text. For example, the term "Theoretical Quantiles" is used without clarification—this concept is not widely familiar and would benefit from a brief explanation of what it represents and why it's used. If space is a concern, one option would be to provide a more detailed description in the caption of Figure 2, and then refer back to it in the captions of subsequent figures. This would balance clarity with conciseness across the manuscript and will aid in understanding important graphs that are perhaps not intuitive.

The theoretical quantile refers to the expected quantiles values under the fitted reference distribution, which are compared with the empirical quantiles of the observed data (i.e., the 1-hr rainfall rate data in the Valencia region). We clarified this point in Fig. 2's caption. Thanks for this recommendation.

A second point relates to your use of the Clausius–Clapeyron (CC) rate (7% per K) as a benchmark throughout the results figures. While this reference makes sense for variables like precipitation and precipitable water, applying it to other variables—such as CAPE, diabatic heating, or maximum graupel—results in a mismatch. These are not expected to scale with CC in the same way, making such comparisons misleading. As shown in Figure 5 of Romps (2016), CAPE begins to diverge from CC scaling above ~ 300 K. This highlights the importance of including temperature information earlier in the manuscript, as noted in my comment on the introduction. Without that context, it's difficult to assess whether CC scaling is even applicable in your case. You also reference Pujol et al. (2021) to support the comparison, but I couldn't find this citation in your reference list, nor locate it independently. Unless there is a strong justification, I would recommend removing the CC reference line for these variables. Instead, comparisons to known or expected climate change responses specific to each variable would be more meaningful and would strengthen the interpretation of your results.

We thank the reviewer for this helpful comment. We agree that applying the Clausius–Clapeyron (CC) rate as a reference to variables such as diabatic heating or maximum graupel may be misleading, as these are not expected to follow CC scaling. With regard to CAPE, and following Pujol et al. (2021), "*CAPE depends on LFC temperature, which is influenced by both surface temperature and surface moisture. However, moisture increases much faster than temperature, at approximately the CC rate. Therefore, CAPE increases could exhibit CC-like scaling with the temperature increase of the LFC*" Nevertheless, in line with the reviewer's suggestion, and acknowledging that it cannot be robustly stated that all these parameters follow CC scaling, we have removed the CC reference lines for these variables in the corresponding figures and text. We also corrected the reference list, as the Pujol et al. (2021) citation had been mistakenly omitted.

Pujol, B., Prien, A. F., Molina, M. J., & Muller, C. (2021). Dynamic and thermodynamic impacts of climate change on organized convection in Alaska. *Climate Dynamics*, 56(7), 2569-2593.

Changes in rainfall intensity and area

L123 The grammar here is a bit awkward. Please remove 'notably' and insert 'well' after 'distribution of precipitation' to improve readability.

Done, thanks.

L124 See the earlier comment on the introduction regarding Figure 1b—its interpretation is currently unclear and may benefit from some redesign for clarity.

Thank you for your suggestion. As stated in the previous comment, we believe that the current Figure 1b version offers the most balanced and informative representation and we think that the westward displacement is well observed.

L127 To ensure consistency in terminology, replace 'present-day' with 'factual'. You could also improve the flow by introducing this analysis with a sentence such as: "When comparing the factual and counterfactual simulations..."

Done. Thank you.

L134–137 The language used here is quite strong considering the considerable uncertainty in the results. It would be helpful to present the full context rather than focusing only on the appealing outcomes. In the figure, the P99 does not appear to add much value—do you need it to support your main argument? If its inclusion is intended to highlight the limitations of your analysis (i.e., that the noise or uncertainty may exceed the signal), that would be a valid point, but please clarify this explicitly in the text. That said, I do appreciate the use of the AEMET data—it provides a valuable link to operational relevance.

It was clarified in the revised manuscript [L157-161]. Thanks.

L140–143 You refer to 'similar studies' (plural) but cite only one. While it's not necessary to elaborate on each study's findings, consider adding two or three relevant references after "... findings from similar studies." You can then remove the following sentence for a more concise presentation.

Thank you for this helpful point. We have removed the following sentence, and added the references that we

forgot to include.

L144–150 This is a strong and thoughtful analysis of quite interesting results. It would be even more compelling if supported by a factual–counterfactual difference map (see earlier comment). I would also recommend avoiding the term ‘violin plot’ here—besides the need for a proper panel label, the figure shows a distribution. It’s better, in my opinion, to refer to the content (e.g., distribution of values) rather than the specific plotting method.

Thank you for your thoughtful feedback. Regarding the suggestion to include a factual–counterfactual difference map, we believe that the differences between both scenarios are already clearly discernible in the current figure. As for your comment on terminology, we have revised the reference to the distribution in the figure and removed the term “violin plot,” as you suggested.

Changes in moisture content and fluxes

L169 Consider starting the paragraph with a reference to Figure 3a. This would guide the reader toward the relevant visual right away, rather than waiting until L172.

Done. Thanks.

L173 You choose for ‘counterfactual’ earlier in the paper, please use consistently and replace ‘pre- industrial’.

Done. Thanks.

L185–186 You might consider replacing ‘In contrast’ with ‘As expected’, though this depends on your interpretation. Given the thermodynamic differences in your experimental setup, such results could be anticipated. I’ll leave the choice to your judgment. Also, I suggest revising ‘notable PW reduction’ to ‘notably lower PW’ for clarity and smoother phrasing.

Thank you for your suggestions. They were amended.

L211–220 This section is an exact duplicate of the section before it, please delete.

Indeed. Thank you.

Changes in physical mechanisms controlling extreme rainfall

L222 The word ‘rainfall’ is uncountable and does not take a plural form. Please remove the ‘s’ at the end of the section title.

Done. Thanks.

L226–227 The sentence is unclear and difficult to follow. Please rewrite.

It was rewritten. Thanks.

L231–234 You might consider referring back to the CAPE results in combination with the updraft analysis. This could help provide a more complete and convincing interpretation of the findings presented here.

Thank you for this point to improve the interpretation of the updraft analysis. We referred to the CAPE results.

L243–255 The argument presented in the graupel paragraph would benefit from additional nuance. While it is well established that deep convection can enhance graupel formation through stronger updrafts and increased riming (e.g., Rutledge and Hobbs (1983); Tao et al. (2012)), the link between higher graupel content and increased rainfall rates is not straightforward. This relationship is influenced by microphysical parameterizations and storm dynamics, and remains highly model-dependent (e.g., Milbrandt and Yau (2005); Morrison et al. (2009)). In some contexts, graupel may enhance precipitation efficiency through melting or collision–coalescence, while in others it may indicate inefficient fallout or hail production. As such, additional justification and context are needed before drawing firm conclusions about the role of graupel in amplifying precipitation rates. The claim that increased graupel combined with stronger updrafts leads to super-Clausius–Clapeyron scaling requires more justification, as current literature shows this outcome is conditional and context-dependent (e.g., Singleton and Toumi (2013); Muller and Takayabu (2020)). Microphysical enhancements like graupel can influence precipitation intensity, but super-CC scaling also depends on storm structure, CAPE, and precipitation efficiency.

We thank the reviewer for this insightful comment to give the readers a context in the relation between graupel

and higher precipitation. We agree that the relationship between graupel content and precipitation intensity is not straightforward and is highly dependent on microphysical parameterizations, storm structure, and precipitation efficiency, as highlighted in the cited literature. In the revised manuscript we have added a context statement [L310–313] to clarify that. Despite stronger updrafts and enhanced graupel can contribute to heavy precipitation through melting and coalescence processes, the role of graupel is context-dependent. We now explicitly acknowledge that in some environments graupel may increase precipitation efficiency, while in others it may indicate inefficient fallout or hail production. Accordingly, we have rephrased our discussion of potential super-Clausius–Clapeyron scaling to emphasize that this outcome is conditional upon storm structure, CAPE, and overall precipitation efficiency, rather than a direct consequence of graupel enhancement.

Discussion

In the discussion section, you bring together the individual variables and interpret them in relation to one another, which is valuable and strengthens your narrative. However, this integrative analysis could be expanded further to deepen the reader’s understanding of how the variables interact within the system.

More importantly, the discussion currently lacks a critical reflection on the limitations of your modeling setup and the nuance or uncertainty associated with your findings. Including a paragraph that openly addresses these limitations—such as model resolution, parameter choices, handling of dynamics, or the representativeness of your results—would improve the transparency and credibility of your conclusions.

Thank you for this point. We agree that the paper needs a limitation statement with a critical reflection on the limitations of our approach. In the Methods section we have added a statement assessing the limitation regarding the modeling setup, PGWA and change per degree warming concern, as you pointed above.

Another important element that could strengthen your discussion is situating your results more explicitly within the context of existing literature. Are your findings consistent with those from previous attribution or PGW studies in the Mediterranean region or other flash flood events? For example, how do your estimates of precipitation amplification compare to similar analyses (e.g., Zappa and Shepherd (2017); Orth et al. (2021))? Explicitly highlighting whether your results confirm, extend, or diverge from these earlier studies would help position your contribution more clearly within the field.

This also provides readers with a sense of how robust or novel the physical mechanisms you identify may be across different events and modeling approaches.

L273–275 Please clarify that your study does not provide evidence for changes on a global scale. I understand that this is not your intention, but the current phrasing could be misinterpreted. Consider rewording to clearly position your findings within the context of a regional study, while linking them to broader global trends in flash flood occurrence.

It was amended following your suggestion. Thanks.

L282–286 This is indeed the appropriate place to synthesize the various variables into a collective interpretation of your results — something you could do even more extensively. That said, this paragraph currently lacks nuance, context, and critical reflection on the limitations of your setup. While it is valid to explore the potential for super-Clausius–Clapeyron (CC) scaling using the variables you’ve analyzed, the paragraph frames this as the primary goal, whereas your results offer more than just evidence for super-CC behavior.

Additionally, you argue that increased WVFlux underscores enhanced moisture transport rather than wind changes, but it is unclear how this conclusion is derived, given the limited description of how dynamics are treated in your model setup. Presumably, the use of upper-atmospheric nudging means that large-scale wind patterns are constrained, allowing you to attribute WVFlux increases primarily to humidity. However, nudging divergence may still influence vertical motion via the continuity equation. It would strengthen your interpretation to discuss how this nudging approach affects updrafts and what this implies for your conclusions.

Finally, your mention of “non-linear processes” returns to the earlier concern about the appropriateness of expressing your results in percent per degree Kelvin (%/K). A discussion of this metric’s limitations — especially when applied to quantities like CAPE or graupel — would be valuable here.

Thank you for this insightful comment, which has improved the manuscript. We agree on the limitations of

exploring the potential for super-CC behavior based on a single case study. However, we would like to emphasize that our primary objective is not to comprehensively assess super-CC scaling, but rather to highlight the sub-daily changes observed in this event and to examine whether it may exhibit characteristics consistent with potential super-CC behavior.

As we noticed above, we did not apply spectral nudging in our simulations due to the lead time being only 36-h, and the domain is strongly constrained by lateral boundary conditions from IFS analysis at 0.1° resolution every 6-h. Therefore, we consider that additional spectral nudging is unnecessary, as it could dampen mesoscale variability. In addition, we added in the manuscript that we separately assessed the changes in the wind and moisture in the low-mid levels to analyze which contributes more to the enhanced WVFlux in the factual simulation. The results show a statistically significant increase (Mann-Whitney U test) in the moisture content in the low-mid levels (Figure RR1g), whereas no significant changes were found in the wind speed field.

Concerning the discussion of scaling, as explained in response to the previous concern, we removed the Clausius-Clapeyron label in some parameters, such as CAPE or graupel. Nevertheless, we consider that non-linear processes occurred in the Valencia event, which promotes the potential super-CC behavior in precipitation, since the relative changes are not uniform across all physical processes. For instance, the increase in WVFlux is notably lower than the increase in latent heat release, i.e., which suggests that greater moisture availability can promote—under adequate conditions—much more intense and active convection. Finally, we have included a limitations statement in the Methods section, where we acknowledge and detail all the caveats of our experimental design and interpretation.

Figure 5 This is an excellent figure that effectively synthesizes your key findings. It deserves more emphasis in the manuscript. I would suggest positioning it as a highlight or centerpiece figure.

Thank you very much for these words. We really appreciate you like Figure 5. We added the magnitude changes in the discussion to emphasize more the figure in the revised manuscript.

Final remark

Taking the feedback into account, I believe this paper will make a strong contribution to the field. It has been an honour to be involved in the process. I am not a fan of single-blind reviewing and will therefore sign my name.

Linda van Garderen

Thank you.

References

- Dare, R. et al. (2019). Attribution of flash flooding events using a storyline approach. *Journal of Hydrometeorology*, 20(6):1234–1246.
- Milbrandt, J. A. and Yau, M. K. (2005). A multimoment bulk microphysics parameterization. part ii: A proposed three-moment closure and scheme description. *Journal of the Atmospheric Sciences*, 62(9):3065–3081.
- Morrison, H., Milbrandt, J. A., Thompson, G., and Bryan, G. (2009). Impact of cloud microphysics on the development of trailing stratiform precipitation in a simulated squall line: Comparison of one- and two-moment schemes. *Monthly Weather Review*, 137(3):991–1007.
- Muller, C. J. and Takayabu, Y. N. (2020). Response of precipitation extremes to warming: What have we learned from theory and idealized cloud-resolving simulations, and what remains to be learned? *Environmental Research Letters*, 15(3):035001.
- Orth, R. et al. (2021). Using storylines to understand the impact of climate change on flash floods in the european alps. *Journal of Hydrometeorology*, 22(3):1233–1245.
- Romps, D. M. (2016). Clausius–clapeyron scaling of cape from analytical solutions to radiative–convective

equilibrium. *Journal of the Atmospheric Sciences*, 73(9):3719–3737.

Rutledge, S. A. and Hobbs, P. V. (1983). The mesoscale and microscale structure and organization of clouds and precipitation in midlatitude cyclones. viii: A model for the “seeder–feeder” process in warm-frontal rainbands. *Journal of the Atmospheric Sciences*, 40(5):1185–1206.

Schär, C., Frei, C., Lüthi, D., and Davies, H. C. (1996). Surrogate climate-change scenarios for regional climate models. *Geophysical Research Letters*, 23(6):669–672.

Shepherd, T. G. (2014). Atmospheric circulation as a source of uncertainty in climate change projections. *Nature Geoscience*, 7(10):703–708.

Shepherd, T. G. (2016). Storyline approach to the construction of regional climate change information. *Proceedings of the Royal Society A*, 472(2186):20160145.

Shepherd, T. G. (2021). Bringing physical reasoning into statistical practice in climate-change science. *Climatic Change*, 169(2):2.

Singleton, A. and Toumi, R. (2013). Super-clausius–clapeyron scaling of rainfall in a model squall line. *Quarterly Journal of the Royal Meteorological Society*, 139(671):334–339.

Tao, W.-K., Simpson, J., and McCumber, M. (2012). An ice-water saturation adjustment. *Meteorology and Atmospheric Physics*, 50(1-3):107–122.

Zappa, G. and Shepherd, T. G. (2017). Storylines of atmospheric circulation change for european regional climate impact assessment. *Journal of Climate*, 30(16):6561–6577.

References cited in the revision:

Brands, S., Gutierrez, J. M., & San-Martin, D. (2017). Twentieth-century atmospheric river activity along the west coasts of Europe and North America: Algorithm formulation, reanalysis uncertainty and links to atmospheric circulation patterns. *Climate Dynamics*, 48(9–10), 2771–2795. <https://doi.org/10.1007/s00382-016-3095-6>

Davolio S., Vercellino M., Miglietta M. M., Drago Pitura L., Laviola S., Levizzani V.: The influence of an atmospheric river on a heavy precipitation event over the Alps, *Weather and Climate Extremes*, 39, 100542, 2023; <https://doi.org/10.1016/j.wace.2022.100542>

Lorente-Plazas R, Montavez JP, Ramos AM, Jerez S, Trigo RM, Jimenez-Guerrero P (2020) Unusual atmospheric-river-like structures coming from Africa induce extreme precipitation over the western Mediterranean Sea. *Journal of Geophysical Research: Atmospheres*, 125, e2019JD031280. <https://doi.org/10.1029/2019JD031280>

Reviewer #1 (Remarks to the Author):

Second review for: Human-induced climate change amplification 1 on storm dynamics in Valencia's 2024 catastrophic flash flood (amended title)

Calvo-Sancho et al.

I'd like to thank the authors for responding comprehensively to my reviewer comments, and in particular for providing a comparison against simulations with a range of lead times in order to provide confidence in the robustness of the results. I have read those parts of the revised manuscript relevant to my comments and am happy with the changes made.

We thank the reviewer for the positive and constructive feedback. We hope our response addresses the concern raised.

I just have one very minor comment which the authors may wish to consider:

L354: Please briefly explain what you infer from the set of simulations with different initialization times (e.g., as you describe in your response doc). Also, I think 'initialization times' would be clearer terminology than 'initializations' (also L353).

Thank you for this point to clarify the results obtained in the different initialization times. We added a sentence as we described in the previous response letter "*All initialization times exhibit similar relative changes between factual and counterfactual runs, which provides confidence that the main conclusions are not sensitive to the specific initialization chosen*" [L357-L359]. Thanks again.

NCOMMS-25-38404

“Human-induced climate change amplification on storm dynamics in Valencia’s 2024 catastrophic flash flood” by Calvo-Sancho et al.

The authors have addressed most of the reviewers' comments, clarifying doubts and amending the text and/or figures to enhance the paper's quality and readability. I appreciate the improved connection to existing literature and for having better emphasized the novel aspects of their work, particularly those related to the sub-daily scale, which I find the most interesting. I have a few more comments for the authors which do not affect my recommendation to the Editor that the article is eligible for publication, as it can contribute significantly to understand the role of global warming in the intensification of extreme precipitations.

Regarding my comment on Figure 1, as I explicitly wrote in the previous review, I did not ask to add any figures or further formal analysis to be shown, but just a summary comment to assess the ability of WRF to reproduce correctly or as fair as possible the real event evolution, which would indirectly add robustness to the conclusions.

We thank the reviewer for clarifying this point. We agree that providing a summary assessment of the factual simulation is essential to support our findings. We have added a summary assessing the WRF model’s capability to resolve the evolution of the event, based on comparisons with rain gauge observations from multiple sources. “*The factual simulation reproduces well the overall spatial distribution of precipitation, although the simulated precipitation field is slightly displaced westward relative to the observations (Figure 1b). The highest amounts of precipitation are concentrated in a central region, aligning relatively well in terms of location with the station-based observation patterns (Figure 1b)*” [L138-140] and “*Although the factual precipitation field is slightly displaced westward relative to the observations, the findings presented in this regional study are consistent with broader evidence that human-induced climate change is intensifying the global hydrological cycle (Fischer and Knutti, 2015; Zhou et al. 2023).*” [L306-309]. Thanks again.

Extended Data Figure 7: I do see that “precipitation efficiency is the ratio Precipitation/Condensation”, the comment was relative to the x-axis tick labels of the PDFs; the clarifications for previous and this figure captions have anyway clarified my concern.

L116 “in a matter of hours” might sound better “within few hours”

Thank you for clarifying the manuscript. We replace it to “within few hours” [L98]

Figure 1 caption: (a) Geopotential height ;

Added. Thanks.

In general I would use the verb “to highlight” instead of “to underscore”

Thank you for this point. We replaced “to underscore” for “to highlight”.

L 238-239: The lack of change in the wind field is in part due on the experiment set-up (as explained in the Limitations section), which does not allow to draw conclusions about their role in the changes of the event respect with pre-industrial climate. On the other hand the set-up itself allow to isolate the role of heat and vapor in modulating the event intensity.

We thank the reviewer for this point. We agree that the lack of significant change in the wind field is in part due to the experiment setup. To ensure this caution is taken into account when interpreting the results, we have added the following sentence: *Nevertheless, it should be noted that the wind components have not been forced by the climate perturbation signal (see Methods section).* [L241-242]

Extended Data Figure 8: I prefer the version of this figure in the reviewers' report, with also differences between factual and counterfactual profiles are plotted.

We thank the reviewer for this point. In the revised manuscript, we completely forget to replace the draft to the final figure. Thanks again.

Second Review: Human-induced climate-change amplification of storm dynamics during the 2024 catastrophic flash flood in Valencia

I thank the authors for their detailed response to my first review and for the revisions that further clarified the manuscript. In my view, the paper is close to being ready for publication.

I only have a few minor comments.

We thank the reviewer for these insightful comments to improve our work.

1) Presentation of difference maps

Thank you for your explanation regarding the figure structure. I fully understand your rationale for organizing the visuals by variable, and I agree that presenting spatial patterns alongside their statistics can support an integrated narrative — this is a reasonable and defensible choice.

That said, even within such a structure, the absence of a difference map still places interpretative effort on the reader. While the differences can indeed be spotted by comparing the factual and counterfactual maps, asking the reader to do this work reduces clarity and weakens the immediacy of your message. A difference map would make the key signal explicit and strengthen the results, rather than requiring the reader to extract it manually.

I do not insist on a specific solution, but I would encourage reconsidering. If space is a concern, one option could be replacing the factual/counterfactual pair with a single difference map. Alternatively, adding the difference map into the existing panel (or as supplementary material) would also solve the issue without changing your narrative structure.

To be clear, this remains a recommendation — the final choice is yours. My view is simply that providing a difference map in some form would make the manuscript stronger and more reader-friendly.

We thank the reviewer for this constructive comment. We agree that difference maps can enhance clarity and make the key signal more explicit for the reader. We added the difference maps in the Supplementary Material in order to increase the robustness of our results and make the interpretation more reader-friendly, as suggested. However, we have decided to place them in the Supplementary Material rather than the main manuscript to maintain the current visual structure and to avoid excessive plotting for each parameter with the main text figures. Thanks again.

2) Lines 122–124: interpretation of Zappa & Shepherd (2017)

The manuscript currently cites Zappa & Shepherd (2017) as supporting increased extremes in Mediterranean winter precipitation. Their findings instead indicate a reduction in Winter precipitation over the Mediterranean under climate change and do not provide evidence for increased extremes. Please revise the text to reflect this and situate your results within that context.

We thank the reviewer for this point to clarify the revised manuscript. We amended the mistake and we situated our results in the context [L122-125]

3) Lines 141–144: vertical mismatch of the storm

Thank you for clarifying in your response that you chose to let the atmosphere freely respond, given the strong lateral boundary conditions. This is a reasonable modeling choice, and the results indeed remain convincing and well-interpretable.

At the same time, this decision likely contributes to the slight vertical shift observed between MSLP and GPH-Z500. Since vertical gradients are central to your analysis, it would be useful to briefly acknowledge how this mismatch arises from the chosen setup and to discuss its potential influence on the results. Although the overall atmospheric dynamics remain comparable, the fields are not perfectly aligned — which may influence the location and intensity of precipitation — and explicitly noting this would help contextualize the interpretation for the reader.

A short paragraph in the Supplementary Material outlining this point (and why, despite it, the comparison remains valid) would be valuable for readers interested in methodological rigor and model behavior. It would also pre-empt questions about the implications of allowing free atmospheric adjustment. In my view, this addition would not complicate the narrative but would strengthen confidence in the approach and transparently frame one of its limitations.

Thank you very much for this insightful comment and for the opportunity to further clarify this point. We fully agree that allowing the atmosphere to evolve freely, instead of applying a spectral nudging, can introduce some degree of vertical mismatch between MSLP and mid-tropospheric geopotential fields. This is a well-known aspect of regional modelling, and we appreciate the reviewer for highlighting it.

For the particular case examined in our study, however, a detailed re-examination of the simulations confirms that the MSLP and Z500 fields exhibit a very high degree of spatial coherence, with structures that closely mirror each other and are fully consistent with the synoptic pattern present in the ERA5 (Figure 1a). The minor displacement noted in Lines 141–144 is also present in the large-scale observations and therefore appears intrinsic of the storm system rather than induced from model decoupling associated with the absence of spectral nudging.

Thank you for your hard work — I look forward to seeing this paper published.

Thank you.

Linda van Garderen